# Multiscale modeling of human cerebrovasculature: A hybrid approach using image-based geometry and a mathematical algorithm

**Satoshi Ii**[1,2]*, **Hiroki Kitade**[2], **Shunichi Ishida**[3], **Yohsuke Imai**[3], **Yoshiyuki Watanabe**[4], **Shigeo Wada**[2]

**1** Graduate School of Systems Design, Tokyo Metropolitan University, Hachioji, Tokyo, Japan, **2** Graduate School of Engineering Science, Osaka University, Toyonaka, Osaka, Japan, **3** Graduate School of Engineering, Kobe University, Kobe, Hyogo, Japan, **4** Graduate School of Medicine, Osaka University, Suita, Osaka, Japan

* sii@tmu.ac.jp

**Data Availability Statement:** All data files are available from the Zenodo database (https://zenodo.org/record/3707179). Visit the URL and follow the citation rule when the data will be used.

## Abstract

The cerebral vasculature has a complex and hierarchical network, ranging from vessels of a few millimeters to superficial cortical vessels with diameters of a few hundred micrometers, and to the microvasculature (arteriole/venule) and capillary beds in the cortex. In standard imaging techniques, it is difficult to segment all vessels in the network, especially in the case of the human brain. This study proposes a hybrid modeling approach that determines these networks by explicitly segmenting the large vessels from medical images and employing a novel vascular generation algorithm. The framework enables vasculatures to be generated at coarse and fine scales for individual arteries and veins with vascular subregions, following the personalized anatomy of the brain and macroscale vasculatures. In this study, the vascular structures of superficial cortical (pial) vessels before they penetrate the cortex are modeled as a mesoscale vasculature. The validity of the present approach is demonstrated through comparisons with partially observed data from existing measurements of the vessel distributions on the brain surface, pathway fractal features, and vascular territories of the major cerebral arteries. Additionally, this validation provides some biological insights: (i) vascular pathways may form to ensure a reasonable supply of blood to the local surface area; (ii) fractal features of vascular pathways are not sensitive to overall and local brain geometries; and (iii) whole pathways connecting the upstream and downstream entire-scale cerebral circulation are highly dependent on the local curvature of the cerebral sulci.

## Author summary

Cerebral autoregulation in the complex vascular networks of the brain is an amazing achievement. We believe that numerical analysis of the cerebral blood circulation using an anatomically precise vascular model provides a powerful tool for evaluating the direct

**Funding:** S. Ii, S. Ishada, YI, and SW were supported by MEXT as "Priority Issue on Post-K computer (Supercomputer Fugaku)" (Integrated Computational Life Science to Support Personalized and Preventive Medicine) (Project IDs: hp150274, hp160218, hp170265, hp180202, hp190187), and S. Ii, S. Ishada, and SW were supported by JSPS KAKENHI grant number JP19H01175. The funders had no role in study design, data collection and analysis, decision to publish, or preparation of the manuscript.

**Competing interests:** The authors have declared that no competing interests exist.

relationships between local- and global-scale blood flows. However, there is a lack of information about the overall vascular pathways in the human brain, preventing a monolithic model of the human cerebrovasculature from being established. This paper presents a multiscale model of human cerebrovasculature based on a hybrid approach that uses image-based geometries and a newly developed mathematical algorithm. One important argument of this paper is the validity of the cerebrovasculature represented in the model, which reflects anatomical features of major cerebral vasculatures and brain shape, and has strong similarities with available data for human superficial cortical vessels. Investigations of the reconstructed model allow us to derive some biological insights and associated hypotheses for the cerebral vasculature. The authors believe the present cerebrovascular model can be applied to numerical simulations of the entire-scale cerebral blood flow.

## Introduction

The cerebral circulation plays an important role in the continuous delivery of oxygen and glucose to brain tissues. An important feature of cerebral circulation is the maintenance of a constant cerebral blood flow (CBF), regardless of mean arterial pressure, known as cerebral autoregulation [1–3]. This involves control of local and global CBF in relation to the metabolic demands from neural activities [4–6]. Understanding how this neuro-vascular coupling affects CBF is essential for achieving an understanding of the core of the cerebral circulation regulation mechanism, and requires the uncovering of physical aspects of the full-scale cerebral blood circulation and complex networks derived from anatomical structures. The cerebral vasculature has a hierarchical structure ranging from vessels of a few millimeters, to superficial cortical vessels with diameters of a few hundred micrometers, to the microvasculature (arteriole/venule) and capillary beds in the cortex. Cerebrovascular morphologies and structures for human cortical and intracortical vessels in arterial and venous systems were explored in [7], which reported many remarkable features for the pathway structure, vessel anastomosis, and differences between the arterial and venous systems. Details of the microvascular features in the human cerebral cortex were later uncovered in [8, 9]. However, as there is a lack of information about the whole-scale vascular pathway, no monolithic model of the human cerebrovasculature has yet been established. A complete cerebrovascular model would enable the computational analysis of the full-scale cerebral circulation, enabling CBF to be evaluated at scales from single vessels to the whole-brain vascular network.

Medical observations can provide real-world configurations of personalized cerebral vasculatures. Recent advances in medical imaging devices enable us to capture both arterial and venous geometries [10, 11], and a combination of different medical observations with state-of-the-art image processing techniques has resulted in an integrated subject-specific brain model [12]. However, the spatial resolution of this model is insufficient to reconstruct the whole-scale vasculature at both local and entire ranges, with the minimum vessel diameter being approximately 1 mm. Recently, an inventive measurement using high-resolution micro-CT imaging of mouse cerebrovasculatures has been reported [13], where the vasculatures at the whole-brain scale were reconstructed down to the microvascular scale. However, such measurements are generally difficult to apply to the human brain.

Because of this limitation of medical observations, mathematical and numerical models based on functional principles have been proposed. One of the more successful models is constrained constructive optimization (CCO), which was originally proposed to reproduce arterial

trees [14] and has several variations [15, 16]. The CCO model updates a tree structure by adding a new vertex to an arbitrary domain, which offers a new terminal end and branches in a step-by-step manner, so that the evaluation function for the vascular pathway and geometry associated with the blood circulation is minimized. The CCO model has been developed to represent the cerebral vasculature both at the macroscale on the brain surface [17] and at the microscale in the brain cortex [18]. As an alternative, several mathematical models for generating vascular pathways have been proposed. In [19], terminal pairs of arterial and venous vasculatures are formed via sequential generation with a tripod junction on a mesh system. In [20], the global optimality of the vascular systems is achieved using a global constructive optimization method developed from the CCO model. An optimized configuration of the vasculatures related to physiological functions has been modeled using a simulated annealing algorithm [21], and this method has recently been applied to cerebral arteries [22]. However, the mathematical approach itself is not guaranteed to reproduce the actual features of the vascular pathway and personal particularities such as the circle of Willis.

In this regard, synthetic or hybrid vascular models have been developed using an observed (actual) morphology and a mathematical algorithm. There are two typical approaches. One employs a data-driven concept that reproduces the statistical features of the vascular morphology evaluated by real-world data [23–27]. Recently, a whole-scale cerebrovascular model of a mouse that reflects the statistical features of the microvascular morphology has been developed and used for blood circulation simulations [28]. Although the data-driven approach is strong when sufficient data are available, it is difficult to apply to cases in which there is less *a priori* morphological information and a patient-specific morphology must be addressed. The second approach uses an image-based geometry, whereby the vascular model is reconstructed according to an observed image [29] or small-scale vasculatures are created from the terminal ends of the image-based large-scale geometry [30]. This approach can easily incorporate a patient-specific (or personalized) geometry into the modeling, and does not require *a priori* statistical information of the vascular morphology. However, to date, no vascular model for the whole-scale human brain has completely addressed the formation of the end connections of the arterial and venous systems, the hierarchical pathway structures, and particularly the morphological features and vessel anastomoses.

This study aims to model a human cerebrovasculature, including both the arterial–venous systems, and explore the vascular pathway and morphology in the model through comparisons with available data. We apply a hybrid approach based on image-based geometries for macroscale vasculatures (and brain hemispheres) and a newly developed mathematical algorithm for mesoscale vasculatures to link from the micro to macroscale systems. This multilevel region-confined (MRC) algorithm is designed to address the image-based vasculature, hierarchical pathways, and pair-wise coupling of the arterial and venous systems by introducing modeling ideas such as multiple roots, multiple levels, and vascular subregions. The goal of the mesoscale modeling is to generate vascular pathways that enable a suitable supply of blood to the brain tissues while minimizing the vascular volume in the generation region. Although this study employs an image-based vasculature extracted from CT images obtained by the authors, the proposed modeling approach can make use of images obtained in other studies. To evaluate the validity of the present model, we perform a numerical experiment that simulates the pial vessels on human hemispheres before the microvascular systems, and which also generates the pathways and geometries of the entire cerebrovasculature. Moreover, we perform an additional example on a geometrically simplified model to investigate the influence of the cortical folding of the brain on the vascular pathway and relevant CBF.

## Methods

### Ethics statement

(Human Subject Research) Institutional Review Board of Osaka University Hospital No. 16496. The data were analyzed anonymously.

### Concept of cerebrovascular modeling

We model a human cerebrovasculature that ranges from millimeter-scale major vessels to micrometer-scale superficial cortical vessels. The vasculature is modeled for the cerebrum, excepting other components such as the brainstem and cerebellum, and does not include the microvascular system inside the cortex. A hybrid approach using image-based geometries and a mathematical algorithm is applied for the modeling. To reflect anatomical features, we first construct image-based models for the macroscale vasculature and brain shape from medical images, and prepare the inputs for the mathematical model. We then construct a mathematical model for the mesoscale vasculature by employing a newly proposed algorithm, in which hierarchical structures are modeled for a coarse-scale vasculature that broadly spreads over the brain surface and a fine-scale vasculature that is confined within a subregion. A conceptual flowchart for the modeling approach is shown in Fig 1.

### Image-based models for macroscale vasculature and brain surface

**Image processing and shape modeling.** Sequential head 4D-CT angiography images of a Japanese subject were used for the reconstruction of large-sized vessels (Institutional Review Board of Osaka University Hospital, No. 16496). The image resolution was approximately 0.5 mm, and the number of frames in a cardiac cycle was 22. Fig 2A shows the sequential images for contrast media streaming from the arterial system to the venous system. Based on these images, we reconstructed the respective vessel geometries using image-processing software (Amira 5.4.2, Visage Imaging, Berlin). By comparing the reconstructed vessels, we selected the vasculatures at frame numbers 5 and 16, which most reasonably represent the morphological features for the arterial and venous systems (Fig 2B). Note that, at this stage, the venous and arterial vasculatures have not become isolated from one another. We then isolated the arterial and venous systems using a semi-automatic method based on a Boolean operation applied to overlapped domains between the vessel geometries at frames 5 and 16 (Fig 2C). Finally, we obtained vascular centerlines and radii from the isolated vessels (Fig 2D). For the image processing, a threshold-based region-growing method was used to segment the vessel regions. This method selects a seed pixel from a region of images in which a main vessel appears in an arbitrary cross-section, and performs 3-D region growing according to the minimum and maximum thresholds set for the gray values of the images. Skeletonization is then performed through the following processes: (a) a map of the distances from the boundaries of the segmented regions is formed; (b) the segmented voxels are thinned based on the maximum value of the distance map; and (c) the centerlines and vessel radii (evaluated from the distance map) are extracted with the thinned voxels. The edge centerlines are given as a set of discrete points with the coordinates and radii, and the edge connectivity (which edge is connected to the others) is determined. Thus, we can evaluate the vessel length and radius and the connectivity of the vascular network. The details are shown in the Amira user guide. Consequently, large-sized vessels with diameters greater than 1 mm were reconstructed. In this example, the right anterior cerebral artery (ACA) was not extracted accurately because of the low image resolution. Therefore, we used the mirror-image structure of the left ACA as the right ACA.

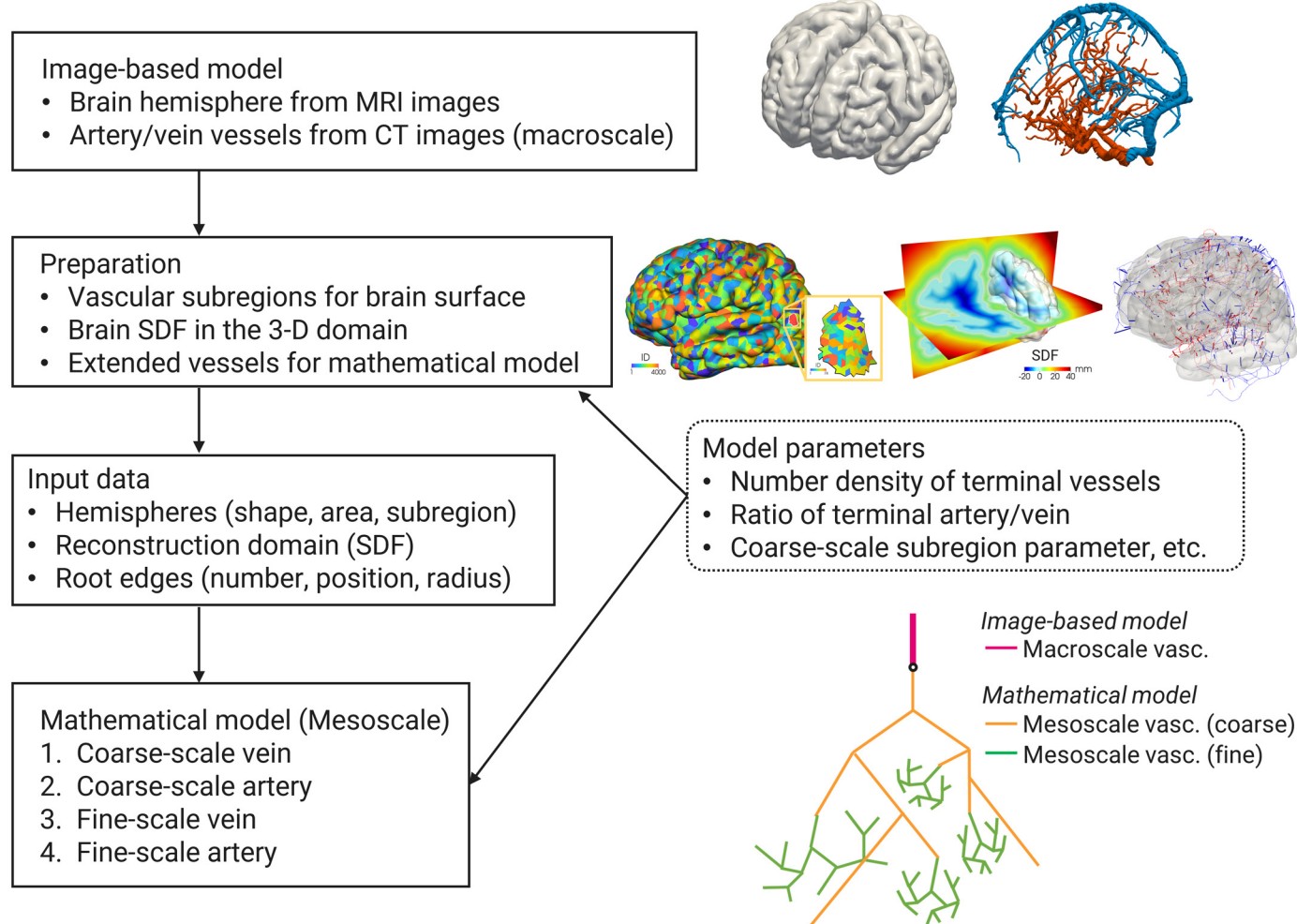

**Fig 1. Concept of the multiscale modeling of cerebrovasculature.** A hybrid approach based on image-based and mathematical models. The brain hemispheres are extracted from MRI images and the macroscale vasculature is reconstructed from CT images. The vascular subregions and signed distance function (SDF) for the brain hemispheres are defined in the preparation process as the inputs for the mathematical model. The mesoscale vasculatures at coarse and fine scales are constructed by a vascular generation (mathematical) algorithm in the multilevel region-confined manner.

The brain hemispheres were reconstructed from MRI images. In this example, the data were acquired from a different subject to the vascular model. The brain surface is reconstructed via image processing performed using Amira 5.4.2, with the surface being represented by a set of polygons (planar triangles). The vascular structure arrangement is then manually adjusted to the brain surface.

**Data preparation for the mathematical modeling of mesoscale vasculatures.** Some terminal ends of the image-based vasculature are located apart from the brain surface. In this paper, we extend each of these to the brain surface with a straight edge. The brain surface $\Gamma$ is implicitly defined by the signed distance function (SDF) $\psi(\mathbf{x})$ in three-dimensional space $\mathbf{x} \in \mathbb{R}^3$, where $\psi$ is discretely given on background Cartesian meshes. Because of the characteristics of the SDF, the distance and direction to the brain surface at any position are given as $\psi$ and $\nabla\psi/|\nabla\psi|$, respectively, through an interpolation. Consequently, the straight edges can be modeled. If the distance of the extended straight edge is longer than a threshold value, the vessel is regarded as not being for the cerebrum, and is neglected in the reconstruction. To introduce the vascular subregions, which are used in the mathematical modeling, the brain surface

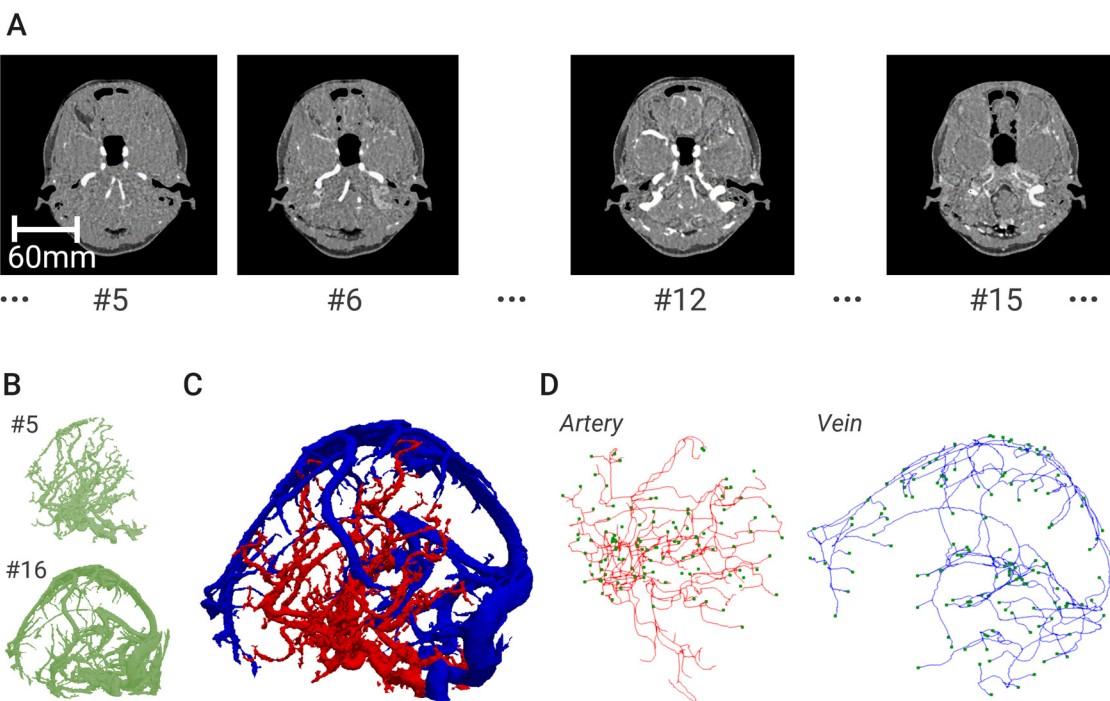

**Fig 2. Reconstruction procedure of the image-based vasculature.** (A) Sequential CT images for bolus injection of contrast media in the brain. (B) Reconstructed vessel geometries at frame numbers 5 and 16, where the venous vasculature is not yet isolated from the arterial one. (C) Isolated vessels for arteries (red) and veins (blue), where the isolation is performed semi-automatically based on a Boolean operation applied to overlapped domains between the vessel geometries at frames 5 and 16. (D) Vascular centerlines (solid line) with radii and terminal points (square symbol) obtained through a thinning process applied to the vessel geometries extracted from (C).

is divided into non-overlapping regions. The seeds of the subregions are randomly distributed to the triangular meshes of the brain surface, and the region growing method is then applied. This procedure was performed for the whole region, followed by each subregion, up to a maximum reconstruction level in the mathematical modeling, i.e., two-level reconstruction in this paper. See the 'Preparation' box and right-hand side figures in Fig 1.

The obtained data for the brain hemispheres (triangular meshes and vascular subregions), reconstruction domain represented by the SDF, and root vessels of the artery and vein (number of roots, position and radius) are utilized in the mathematical modeling as input data.

### Mathematical modeling of mesoscale vasculatures

**Multilevel Region-Confined (MRC) algorithm.** A key issue for modeling the cerebrovascular structure is the reproduction of a hierarchical pathway with connection units for arterial and venous terminals that can cover the surface of the cerebral cortex without excess or deficiency. To accomplish this, we propose a novel vascular generation framework termed the *Multilevel Region-Confined* (MRC) algorithm. In MRC generation, a tree-based vascular generation algorithm creates a set of vascular structures in an arbitrary domain from each root edge so that several terminal ends share the same subregion and form a vascular unit (known as a venous unit in a previous work [7]), which we refer to as a *vascular subregion* in this paper. Fig 3A shows a schematic of the multilevel region-confined manner. Here, three vascular trees are created in the domain $\Omega$, which has vascular subregions $D_i$ ($i = 1, \ldots, N_d$) at the first level (LV1), and further vascular trees are generated in each subregion at the second level (LV2). Potentially, the MRC generation process has the following features:

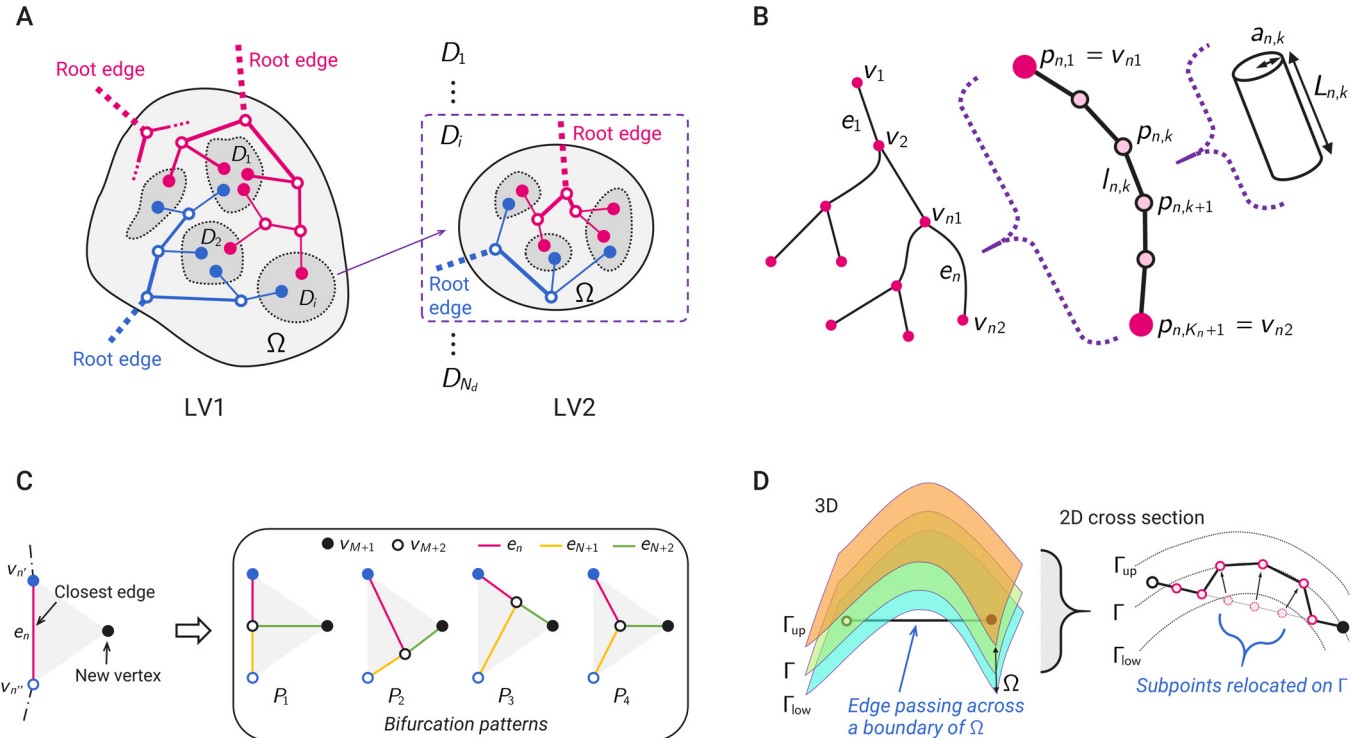

**Fig 3. Schematic of the MRC algorithm.** (A) Vascular generation in the multilevel region-confined manner. In the LV1 reconstruction, two arterial structures (red) and one venous (blue) structure are generated from their root edges, where the terminal vertices enter into respective vascular subregions $D_i$. The following arterial and venous structures are then generated while being confined within each subregion $D_i$ in the LV2 reconstruction. (B) Definitions of the vascular structure. A single graph is shown with vertices $v_m$ and edges $e_n$. Also, a schematic of the edge subdivision for $e_n$ is shown, where the edge is divided by the line segments $l_{n,k}$ with subpoints $p_{n,k}$. In the representation, the edgewise line segment is given as a straight cylinder with the radius $a_{n,k}$ and length $L_{n,k}$. (C) Description of the generation of new branches. Four bifurcation patterns $(P_1, P_2, P_3, P_4)$ are considered for solving the optimization problem (7) when adding a new terminal vertex $v_{M+1}$ to the graph with vertices $v_m$ and edges $e_n$. In the schematic, the edge $e_n$ is selected as the closest edge of the new vertex, and a branching vertex $v_{M+2}$ and edges $e_{N+1}, e_{N+2}$ are newly added. (D) Schematic of the relocation of an edge. Here, the vascular subregions are given on a curved surface $\Gamma$, and the reconstruction domain $\Omega$ is defined as the extended domain within the lower limit surface $\Gamma_{low}$ and upper limit surface $\Gamma_{up}$. As the edge passes across the boundary of $\Omega$ (outside of the lower limit $\Gamma_{low}$), the edgewise subpoints outside the domain are relocated on the curved surface $\Gamma$.

- A hierarchical structure consisting of different scale vascular systems is reproduced in the multilevel process.

- Local vessel densities are controlled by changing parameters related to the vascular subregion (e.g., location, size, number of terminal ends per region).

- Coupling of vascular systems is addressed through the vascular units from different root edges (for arteries and veins).

- Vessel anastomoses at different scales are systematically addressed, making use of the vascular subregion in each generation level.

**Definition of vascular pathways and structures.**    We introduce general definitions of the vascular structures based on descriptions of graph theory. Let $\mathcal{G}(\mathcal{V}, \mathcal{E})$ denote a graph structure with sets of vertices (or nodes) $\mathcal{V}$ and edges $\mathcal{E}$:

$$\mathcal{V} = \{v_m \mid m = 1, 2, \ldots M\}, \tag{1}$$

$$\mathcal{E} = \{e_n \mid n = 1, 2, \ldots N\}, \tag{2}$$

where $v_m$ and $e_n$ are their elements. Here, $M = |\mathcal{V}|$ and $N = |\mathcal{E}|$ are the number of vertices and edges, respectively. It is possible to give each edge as a curved line with an arbitrary function using several degrees-of-freedom (DOFs). Let $\mathbf{p}(s;K)$ denote the edge position in the parametric representation with the parameter (or local coordinate) $s \in [0, 1]$, where $K$ denotes the number of DOFs to represent the edge. The edge length can then be given as $\int_0^1 |\partial \mathbf{p}/\partial s| ds$.

In this study, we simply apply a piecewise linear polynomial to represent the curved edge. Fig 3B shows an example of the vascular pathway and geometry for a single graph $\mathcal{G}$. The edge $e_n$ linking to two vertices $\{v_{n1}, v_{n2}\} \in \mathcal{V}$ is divided into non-overlapping line segments $l_{n,k}$ ($k = 1, 2, \ldots, K_n$), namely,

$$e_n = \{l_{n,k} \mid k = 1, 2, \ldots, K_n\} = \sum_{k=1}^{K_n} l_{n,k}, \tag{3}$$

indicating the edge $e_n$, which discretely represents a curved line with a combination of line segments $l_{n,k}$. Here, the edgewise subpoints are defined as $p_{n,k}$ ($k = 1, 2, \ldots, K_n + 1$), and the line segment consists of sequential subpoints, i.e., $l_{n,k} = \{p_{n,k}, p_{n,k+1}\}$. Note that $p_{n,1} = v_{n1}$ and $p_{n,K_n+1} = v_{n2}$. The edge length $L_n$ is evaluated by

$$L_n = \sum_{k=1}^{K_n} L_{n,k} = \sum_{k=1}^{K_n} \| \overrightarrow{p_{n,k}\ p_{n,k+1}} \|, \tag{4}$$

where $L_{n,k}$ is the length of the line segment $l_{n,k}$. To take account of the vascular thickness, the line segment is assumed to be a cylinder with known radius. In this study, the edge radius $a_n$ is simply defined as

$$a_n = \frac{1}{K_n} \sum_{k=1}^{K_n} a_{n,k}, \tag{5}$$

where $a_{n,k}$ is the cylinder radius of the line segment $l_{n,k}$.

**Geometry-prioritized CCO model.** We apply a simplified and geometry-prioritized CCO model for vascular tree generation, which is developed from the original model [14]. The CCO model updates a tree structure by adding a new vertex to an arbitrary domain that offers a new terminal end and branches in a step-by-step manner. In the original CCO model, both local and global optimization problems are solved to minimize the total intravascular volume: local optimization termed *geometric optimization* to find a bifurcation point, and global optimization termed *structural optimization* to find a connecting edge. However, the geometric optimization requires a nonlinear optimization algorithm with some nonlinear constraints, and moreover, the structural optimization requires multiple evaluations for neighboring existing edges, inferring time consuming tree development involving a massive generation level such as a whole-scale cerebrovascular structure. As the edge closest to the new terminal is the most probable candidate for a permanent connection ($\sim 60\%$) [14], we choose the closest edge to the new terminal as the candidate for the connection edge, without considering the *structural optimization*.

Because of the above reasons, we only consider the *geometric optimization* for generation of a new bifurcation, and this is also much simplified by a combinatorial optimization. We introduce multiple vascular trees with their own roots as $\mathcal{G}_j$ ($j = 1, 2, \ldots, N_0$) and vascular subregions $D_i$ ($i = 1, 2, \ldots, N_d$), where $N_0$ is the number of tree sets or roots and $N_d$ the number of vascular subregions. Let us suppose a new terminal vertex is added in any vascular subregion $D_i$ and the closest edge, which is the minimum Euclidean distance between the terminal vertex and middle point of each existing edge, is found in the graph $\mathcal{G}_j$. Here, two constraints are

imposed: (1) the distance to the closest edge is less than a threshold length $l_{th}$ and (2) the number of terminal ends in subregion $D_i$ is within the permissible range of the number of terminal ends per subregion. If these constraints are not satisfied, the added vertex is rejected; a new terminal vertex with a different subregion and position is then added and re-evaluated until these constraints are satisfied. We define the accepted terminal vertex $v_{M+1}$ and the closest edge $e_n$ that generates a new branching vertex $v_{M+2}$ and edges $e_{N+1}$, $e_{N+2}$ (Fig 3C). As the combinatorial optimization, we consider four bifurcation patterns $P_1$, $P_2$, $P_3$ and $P_4$, in which the new branching vertex $v_{M+2}$ is located on the mid-point of each tentative edge ($P_1$, $P_2$, $P_3$) or centroid of the bifurcation plane formed by three vertices ($P_4$). We choose a bifurcation pattern that minimizes the evaluation function related to the intravascular volume defined as

$$E = \sum_{n'=1}^{N+2} L_{n'} a_{n'}^2. \tag{6}$$

The distribution of radii is evaluated by introducing a flow network model, which will be described later in this paper. It should be noted that as the closest edge is chosen as the connecting edge in this study, the difference in the evaluation function (6) is only attributed to the length and radius for edges $e_n$, $e_{N+1}$ and $e_{N+2}$. Thus, the *geometric optimization* can be written as

$$\begin{aligned} &\text{Find} \ \ P_i \ \ (i = 1, 2, 3, 4), \\ &\text{such that} \ \ \min E' = L_n a_n^2 + L_{N+1} a_{N+1}^2 + L_{N+2} a_{N+2}^2. \end{aligned} \tag{7}$$

The terminal vertex is added to the subregion $D_i$; however, it is not guaranteed that the bifurcation edges in all patterns are located within the reconstruction domain $\Omega$. As shown in Fig 3D, if the bifurcation edges pass across the domain boundary, the edgewise subpoints outside the domain are arranged so that they are located in an arbitrary region $\Gamma \subseteq \Omega$. These modified edges for $e_n$, $e_{N+1}$ and $e_{N+2}$ are taken into account in all the bifurcation patterns when solving the optimization problem (7). Consequently, the graph $\mathcal{G}_j(\mathcal{V}_j, \mathcal{E}_j)$ is updated, where $|\mathcal{V}_j| = M + 1$ and $|\mathcal{E}_j| = N + 2$. The tree generation continues until the number of terminal edges reaches a preset value.

Here, we describe how the edge radii in the mathematical model are determined. The edge radii are determined according to some physical and physiological assumptions. In this regard, although we temporarily introduce an inlet flowrate and a flow network model, the final formulation eliminates the explicit description using the flowrate, and instead only includes a pathway topology and geometric information for the subregion size and root-edge radius. Hereafter, we define the variables for each graph and vascular subregion with the superscripts $(\mathcal{G}_j)$ and $(D_j)$, respectively. Introducing the inlet flowrate of the root edge $Q_{root}^{(\mathcal{G}_j)}$ and the subregion size $|D_i|$ (e.g., line, area or volume depending on the dimension of the subregion), the total flowrate into the domain $\Omega$ is given by

$$Q_{total} = \sum_{j=1}^{N_0} Q_{root}^{(\mathcal{G}_j)}, \tag{8}$$

and the total subregion size is given by

$$|D_{\text{total}}| = \sum_{i=1}^{N_d} |D_i|. \tag{9}$$

In this study, we assume that the total inflow is distributed along the vascular pathways according to the size ratio of the subregion to the total region. The flowrate entering into the subregion $D_i$ is fixed as

$$q^{(D_i)} = \frac{|D_i|}{|D_{\text{total}}|} Q_{\text{total}}. \tag{10}$$

If the subregion has several terminal ends, the flowrate for each terminal edge is assumed to be equally divided by the number of terminal ends per subregion $D_i$, $N_{\text{term}}^{(D_i)}$. Thus, for the graph $\mathcal{G}_j$, the flowrate on a terminal edge $e_{\text{term}}^{(\mathcal{G}_j)}$ entering into an arbitrary vascular subregion $D_{i'}$ is given by

$$Q_{\text{term}}^{(\mathcal{G}_j)} = \frac{q^{(D_{i'})}}{N_{\text{term}}^{(D_{i'})}}, \quad \text{for } e_{\text{term}}^{(\mathcal{G}_j)} \in D_{i'}. \tag{11}$$

For the graph $\mathcal{G}_j$, the internal flowrate on an arbitrary edge $e_n^{(\mathcal{G}_j)}$ is evaluated by a summation of the terminal flowrates of its downstream pathway according to the mass conservation law. Supposing the terminal edges of the downstream pathways for $e_n^{(\mathcal{G}_j)}$ enter into single or multiple subregions $D_{i'}(e_n^{(\mathcal{G}_j)})$ $(i' = 1, 2, \ldots, N_d(e_n^{(\mathcal{G}_j)}))$, where $N_d(e_n^{(\mathcal{G}_j)})$ is the number of downstream terminal edges for $e_n^{(\mathcal{G}_j)}$, the internal flowrate $Q_n^{(\mathcal{G}_j)}$ is given by

$$Q_n^{(\mathcal{G}_j)} = \frac{1}{|D_{\text{total}}|} \underbrace{\sum_{i'=1}^{N_d(e_n^{(\mathcal{G}_j)})} \frac{|D_{i'}(e_n^{(\mathcal{G}_j)})|}{N_{\text{term}}^{(D_{i'}(e_n^{(\mathcal{G}_j)}))}}}_{=S_n^{(\mathcal{G}_j)}} Q_{\text{total}}, \tag{12}$$

where the relation (11) with (10) is used. The edgewise parameter $S_n^{(\mathcal{G}_j)}$ denotes the total size of the apparent coverage subregions of the edge $e_n^{(\mathcal{G}_j)}$. Assuming that Poiseuille flow is established in each cylindrical edge, the wall shear stress exerted on the cylinder wall is given by

$$\tau_n^{(\mathcal{G}_j)} = \frac{4\mu Q_n^{(\mathcal{G}_j)}}{\pi (a_n^{(\mathcal{G}_j)})^3}, \tag{13}$$

where $\mu$ is the fluid viscosity. We assume that the flow is distributed to each edge so that it makes all the wall shear stresses equivalent [31]. Thus, the following relation holds:

$$\tau_{\text{root}}^{(\mathcal{G}_j)} = \tau_n^{(\mathcal{G}_j)}, \quad \Rightarrow \quad \frac{4\mu Q_{\text{root}}^{(\mathcal{G}_j)}}{\pi (a_{\text{root}}^{(\mathcal{G}_j)})^3} = \frac{4\mu Q_n^{(\mathcal{G}_j)}}{\pi (a_n^{(\mathcal{G}_j)})^3}. \tag{14}$$

Consequently, the edge radii are uniquely determined by using the edgewise subregional parameters, $S_n^{(\mathcal{G}_j)}$ and $S_{\text{root}}^{(\mathcal{G}_j)}$, as follows.

$$a_n^{(\mathcal{G}_j)} = \left(\frac{Q_n^{(\mathcal{G}_j)}}{Q_{\text{root}}^{(\mathcal{G}_j)}}\right)^{\frac{1}{3}} a_{\text{root}}^{(\mathcal{G}_j)} = \left(\frac{S_n^{(\mathcal{G}_j)}}{S_{\text{root}}^{(\mathcal{G}_j)}}\right)^{\frac{1}{3}} a_{\text{root}}^{(\mathcal{G}_j)}, \tag{15}$$

for $n \in [1, N^{(\mathcal{G}_j)}]$ and $j \in [1, N_0]$.

Note that the original CCO model determines the edge radii using the pathway tree and the inlet and outlet flow pressures (or flowrates) through a flow network model with a bifurcation rule for the radii of parent and daughter vessels. Thus, the root-edge size is not reflected in the modeling. In our model, the root-vessel sizes obtained by the LV0 model (image-based vasculature) can be incorporated into the modeling, and the edge radii in the model are not dependent on the inflow condition (i.e., total flowrate $Q_{\text{total}}$) but are dependent on the size and number of the terminal ends of the vascular subregions for the pathway structure. Therefore, we have termed the present modified version the *geometry-prioritized CCO model*. In this study, as a main concern was to develop the framework for the cerebrovascular modeling, a strong constraint for the equivalence of the wall shear stress was employed to easily calculate the edge radii (15). As further developments, an alternative approach based on the power law of the bifurcation rule could be considered, to evaluate the edge radii employed in the original CCO model.

We investigated the validity of the present modification through a numerical experiment performed using the original CCO model [14] in S1 Supplement. This confirmed that the present *geometry-prioritized CCO model* is in good agreement with the experiments and original CCO model, indicating that the present modification can be accepted.

**Multilevel reconstruction.** The abovementioned vascular generation is performed at each level of the reconstruction. The information about the vascular geometries of terminal edges (i.e., position and radius) entering the subregion in the lower reconstruction level is inherited from the higher level as the root-edge geometries in the reconstructed domain. In the present formulation, further reconstruction in each vascular subregion is independently performed, as shown in Fig 3A. Thus, the first-level vasculature broadly spreads over the entire domain, whereas higher-level vasculatures are confined to each subregion, which depends on the given vascular subregions and multiple roots in each reconstruction level. The generation continues until the number of terminal edges reaches a preset value.

**MRC generation using the inputs from the image-based model.** For a human cerebrovasculature, the mathematical model is constructed by the MRC algorithm using the inputs extracted from the image-based model. Under the MRC strategy, the model is continuously connected from the terminal edges of the LV0 model (image-based vasculature). The MRC generation is sequentially applied to all arterial and venous vascular systems in each coarse-scale (LV1) and fine-scale (LV2) vasculature. In the modeling, the difference between arteries and veins is attributed to the input roots from the LV0 model and the number of terminal vessels, which is a model parameter determined from measurement data. The flowchart in Fig 4 shows the modeling processes in each level, which can be described as follows.

**Coarse-scale vascular structures (LV1).** The coarse-scale vascular structures are generated for both arteries and veins. The location and geometries of the root edges are inherited from the terminal ends of the LV0 model. New terminal vertices are added through the following process: first, a vascular subregion is randomly selected, then a triangular mesh in the subregion is randomly selected, and finally, a position is randomly determined in the triangular mesh. First, the venous structures are constructed such that all the subregions have at least one

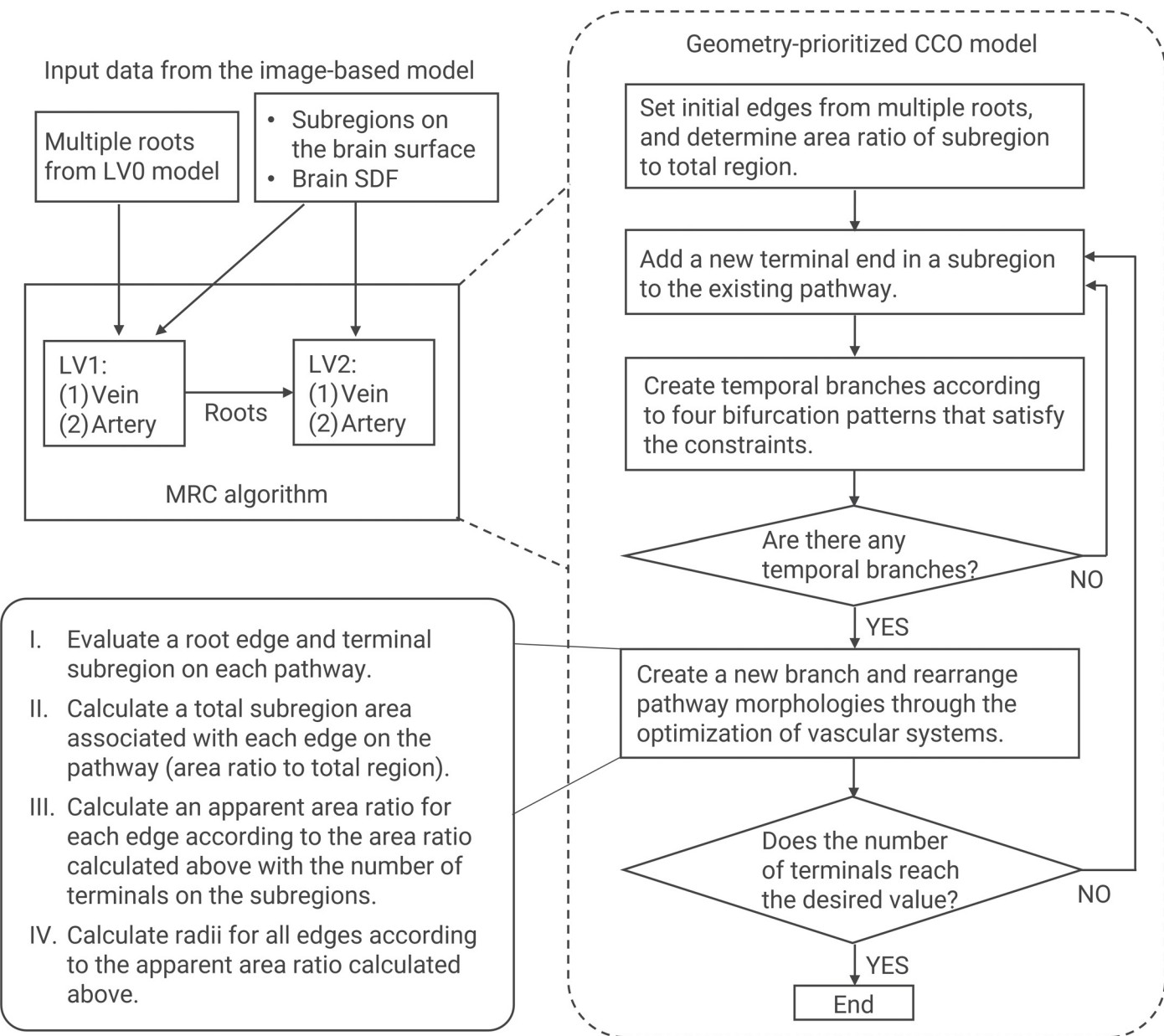

**Fig 4. Flowchart of vascular generation in the MRC algorithm.** The coarse-scale vasculatures for both arteries and veins are constructed in the LV1 reconstruction, where the location and geometries of the root edges are inherited from the terminal ends of the LV0 model. The fine-scale vasculatures are then generated in the LV2 reconstruction, where the generation continues from the terminal ends of the arteries and veins in the former level (LV1). The vascular subregions and SDF for the brain surface are used for the reconstructions at each level. The vascular generation algorithm at each level follows the geometry-prioritized CCO model described in this paper.

terminal end. Next, the arterial structures are constructed using the same set of vascular subregions. With the aim of creating vascular variation, the numbers of terminal arteries and veins per vascular subregion are defined as controllable parameters. To confine the vascular pathway to the brain surface, a discrete SDF $\psi(\mathbf{x})$ is again utilized. The edgewise subpoints outside the domain $\Omega$ are arranged so as to be located on the brain surface $\Gamma = \{\mathbf{x} \mid \psi(\mathbf{x}) = 0\}$. As described in Fig 3D, the reconstruction domain $\Omega$ is defined as the extended domain within the lower

limit $\Gamma_{\text{low}}$ and upper limit $\Gamma_{\text{up}}$, with these being defined as $\Gamma_{\text{low}} = \{\mathbf{x} \mid \psi(\mathbf{x}) = -\psi_{\text{th}}\}$, $\Gamma_{\text{up}} = \{\mathbf{x} \mid \psi(\mathbf{x}) = \psi_{\text{th}}\}$ and $\Omega = \{\mathbf{x} \mid -\psi_{\text{th}} \leq \psi(\mathbf{x}) \leq \psi_{\text{th}}\}$, where $\psi_{\text{th}}$ is the length of the extended domain. If the SDF at subpoints $\mathbf{x}_{n,k}$ becomes $|\psi(\mathbf{x}_{n,k})| > \psi_{\text{th}}$, the related subpoints are moved to the brain surface $\Gamma$ through the interpolation of $\psi$. A similar idea using the SDF for the vascular generation can be seen in [17], where it was combined with the staged-growth CCO model [15].

**Fine-scale vascular structures (LV2).**    The fine-scale vascular structures are independently generated within each LV1-subregion, with the pial vessels being reproduced before they penetrate the cerebral cortex. The generation is continued from the terminal ends of the arteries and veins in the former level (LV1). The number of vascular subregions for the level $N_d^{(LV2)}$ is set to the same value as that for the LV1-subregions. Analogous to the LV1 process, we enforce each fine-scale vascular subregion so that it has at least one terminal end for both an artery and vein.

## Evaluation metrics for mesoscale vasculatures

**Diameter distribution for terminal vessels.**    The data for the terminal edge diameters were collected from all arteries and veins in each LV1 and LV2 reconstruction. The box-whisker plots presented here are based on the collected data. The diameter distributions given in the box-whisker plots were compared with observed data for human cortical and intracortical vessels [7]. Note that the measurement data are characterized using the minimum and maximum diameters for different vessel-type groups. Here, the present data for terminal vessels in the LV1 model are compared with the measurement data for the central and peripheral vessels of the superficial cortical vessels, whereas those in the LV2 model are compared with those for the intracortical vessels before penetration into the cerebral cortex.

**Diameter-defined Strahler order.**    To evaluate the vascular pathways, we introduce the diameter-defined Strahler orders [32]. In this ordering, the terminal edges are assigned an order of 0. The remaining edges are then iteratively ordered, with statistical information about the diameter in each order used as an additive rule to the standard Strahler order. After assigning the orders, continuous edges with the same order are defined as *elements*. For further details, readers are referred to the original paper [32]. To apply this ordering to our reconstructed models, we first connected the LV2 model to the LV1 model for arterial and venous systems, respectively. We then assigned the ordering to each connected arterial and venous vasculature. In the original ordering process, capillary vessels were assigned an order of 0. However, as our approach does not model the capillaries, we set the order of all terminal edges to 1. In this study, we independently evaluate the arterial and venous systems.

**Morphological structures along vascular pathways.**    As morphological structures along vascular pathways related to the abovementioned diameter-defined Strahler order, the diameter $d$, length $l$, and number of elements $N_{elem}$ were collected for each *element* order. Their means and standard deviations are evaluated. We also evaluated regression lines with the form $\log_{10} q = a + bn$, where $n$ is the element order, $q$ is either the diameter, length, or number of elements ($q = d, l, N_{elem}$), and $a, b$ are the intercept and slope for the corresponding variables. These regression lines were also used for quantitative comparisons with the available data for rat (intra-)cortical vessels [33].

**Configuration of vascular whole-pathways.**    To investigate the 3-D configurations of vascular whole-pathways, we calculated the Euclidean distance $L$ and path length $C$ of the whole pathways between the root vertex and terminal ends. The path length $C$ is the summation of the length of all edges in the whole pathway from the root to the terminal edges. We also introduce an index of the whole-pathway tortuosity, given as $C/L - 1$. The evaluation was performed for sequential vasculatures given by the LV1 and LV2 models. The relationships

between *L* and *C* are shown in the form of scatter plots, and the respective distributions are shown as histograms.

**Vascular territories of the major cerebral arteries.** In our modeling, it is easy to identify which vascular subregion has which pathway (end) and the associated root cerebral artery. Based on this information, we determined the vascular territories on the brain surface occupied by the ACA, middle cerebral artery (MCA), and posterior cerebral artery (PCA). We also evaluated each territorial area by summing the area of the vascular subregions.

## Results/Discussion

### Model parameters

The model parameters are summarized in Table 1. The number-density of the terminal vessels over the brain surface area [34] and the artery-vein ratio [9, 27] are obtained from the observational data. However, there are no clear data for the other parameters, and we therefore introduce some assumptions. We set the total number of vascular subregions to be the same as the number of terminal veins from the standpoint of the blood flow supply to the brain surface and associated cortical tissue. As it is hard to determine the ratio between the coarse and fine-scale vascular structures in advance, we first show a model reconstructed under one parameter for the number of coarse-scale vascular subregions $N_d^{(LV1)}$, then move on to discuss the influences of the parameter on the vascular structures.

**Table 1. Mathematical model parameters.**

| Parameter | Attribute | Value |
|---|---|---|
| Surface area of hemispheres, $|D_{\text{total}}|$ | | 102086 mm$^2$ |
| Number of input roots, $N_0$ | artery, vein | 129, 121 |
| Number of total terminal edges, $N_{\text{term}}^{(total)}$ | artery & vein | 888000$^*$A |
| Ratio of total terminal arteries to veins, $\alpha$ | artery/vein | 2$^*$B |
| Number of total terminal arteries and veins | artery, vein | $\frac{\alpha}{\alpha+1}N_{\text{term}}^{(total)}, \frac{1}{\alpha+1}N_{\text{term}}^{(total)*}$B |
| Number of total vascular subregions, $N_d^{(total)}$ | | $\frac{1}{\alpha+1}N_{\text{term}}^{(total)*}$C |
| (LV1) | | |
| –Number of vascular subregions, $N_d^{(LV1)}$ | | 4000$^*$D |
| –Number of terminal edges | artery, vein | $\alpha N_d^{(LV1)}, N_d^{(LV1)}$ |
| –Permitted number of terminals per subregion | artery, vein | $1 \sim 6$, 1 |
| (LV2 per LV1 subregion) | | |
| –Number of vascular subregions, $N_d^{(LV2)}$ | | $N_d^{(total)}/N_d^{(LV1)}$ |
| –Number of terminal edges | artery, vein | $\alpha N_d^{(LV2)}, N_d^{(LV2)}$ |
| –Permitted number of terminals per subregion | artery, vein | 2, 1 |
| Length of the extended domain, $\psi_{\text{th}}$ | | 1 mm |
| Threshold length to the nearest edge, $l_{\text{th}}$ | | 20 mm |

$^*$A We refer to the number density of major penetrating vessels (8.7 num/mm$^2$ [34]) and regard it as the present number density of the terminal edges for the pial vessels before they penetrate the cerebral cortex.

$^*$B The numbers of terminal arteries and veins are determined so that the ratio between them is 2:1 [9, 27].

$^*$C The total number of subregions is set to be the same as the number of terminal veins from the standpoint of the blood flow to the brain surface and associated cortical tissue.

$^*$D This parameter determines the ratio between the coarse- and fine-scale vascular structures.

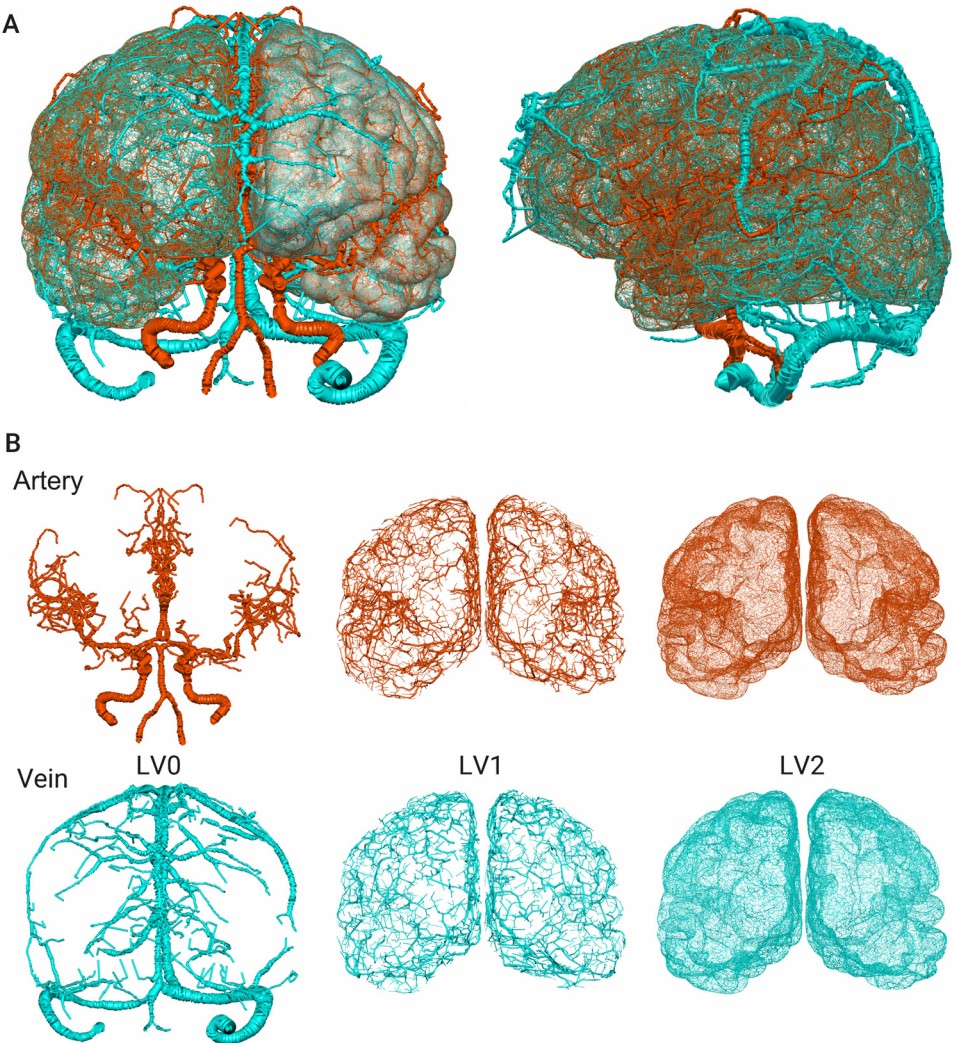

**Fig 5. The reconstructed model for arteries (red) and veins (blue).** (A) Superimposed views of the image-based reconstruction (LV0) and mathematical generation (LV1 and LV2). (B) Respective vascular structures in the different levels.

## Model overviews

A reconstructed model is shown in Fig 5. The model forms a complex pathway spreading over the cerebral surface with changing vessel diameters, with finer vascular structures being observable in higher levels. In the present MRC modeling, as the vascular pathways are generated while confined within the vascular subregion of the brain surface, a set of terminal ends of arteries and veins reaches the same subregion in LV1, which we regard as having formed a vascular unit. Fig 6 visualizes a set of vascular structures reaching the same subregion. Here, there are two arterial pathways and one venous pathway entering the subregion from different root vessels (terminal branches in LV0). As expected, a hierarchical pathway from broad region to local is constructed with respect to the vascular subregion.

The complex cerebrovascular structure is reproduced by the proposed framework. Particular geometric features such as the circle of Willis and the following major cerebral arteries, and the superior/inferior sagittal sinuses and the following cerebral veins, are captured using the

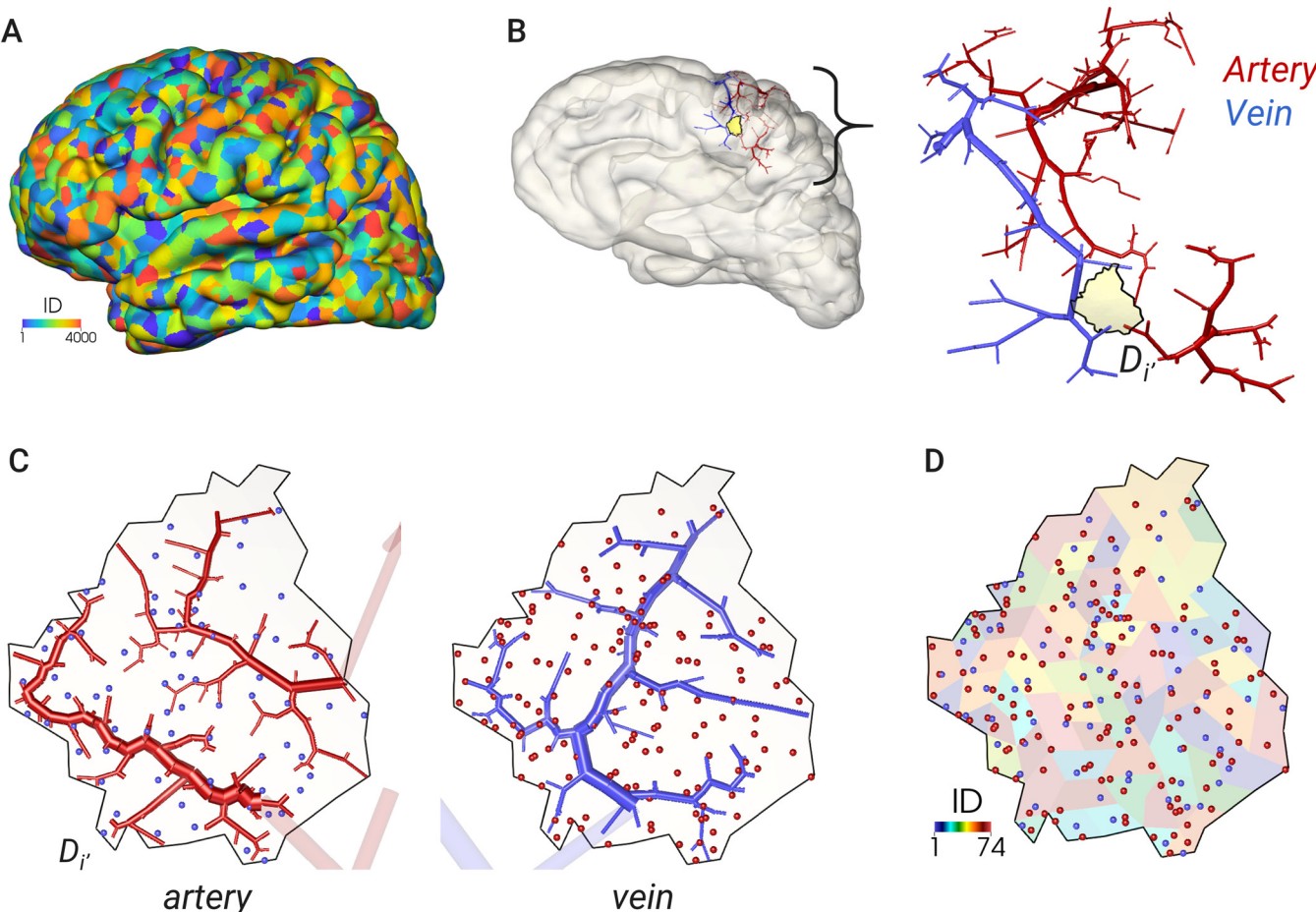

**Fig 6. Vascular pathways to/in a single vascular subregion.** (A) Configuration of the vascular subregion on the brain surface in LV1, with the color indicating the subregion number (ID). (B) Vascular structures in the LV1 reconstruction. Here, there are two arterial pathways and one venous pathway entering an arbitrary vascular subregion $D_{i'}$. (C) LV2 reconstruction within the above subregion $D_{i'}$. The vascular pathways have continued from previous terminal vertices. In each figure for an artery/vein, the terminal ends of another vessel are shown as dots. (D) Configurations of terminal ends for arterial (red) and venous (blue) pathways to the introduced subregions on the brain surface. Note that, owing to modeling constraints, all subregions in LV2 have two arterial ends and one venous end.

medical images in the LV0 process down to a diameter of $\mathcal{O}(1$ mm$)$. The hierarchical vascular structure is then addressed by the MRC generation, where the central and peripheral branches down to a diameter of $\mathcal{O}(100$ μm$)$ are modeled in the LV1 process, followed by the confined network structures of the pial vessels with a diameter around or less than 100 μm being modeled in the LV2 process. The proposed multilevel reconstruction enables adjustment of a ratio between brief and detailed geometric features of the vascular pathway. The terminal ends of the pathways are always located in any vascular subregion that brings a success of forming a pair candidate of arterial and venous systems through a (micro-)vascular unit in the cerebral cortex.

## Superficial cortical vessels

Fig 7 shows an overview and enlarged view of the gyral surface for the LV1 and LV2 vessels given by the present model. The diameters of the terminal edges in the present LV2 model are also shown alongside measurement data for the major intracortical vessels [7]. The present

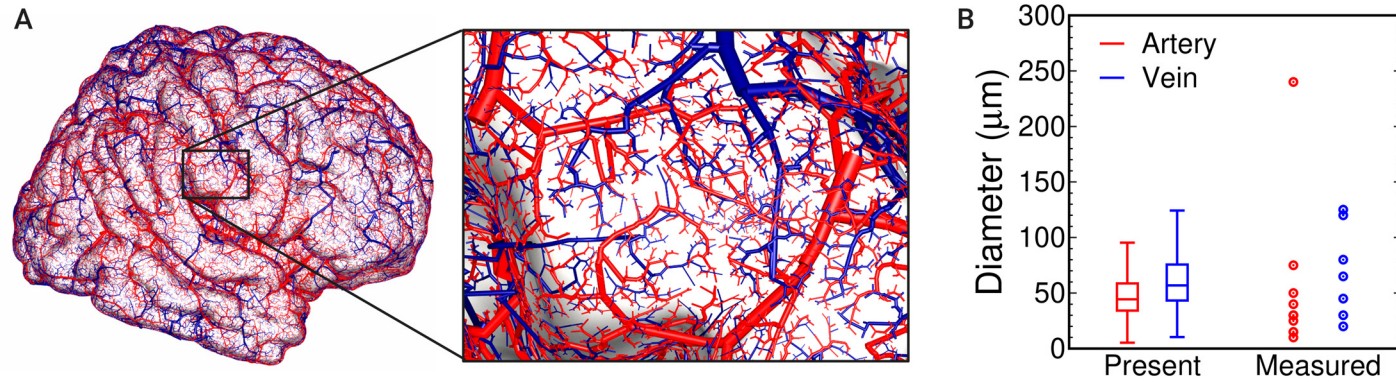

**Fig 7. Comparison of superficial cortical vessels.** (A) Overview of the vascular structure in the present model for the LV1 and LV2 processes and enlarged view of the present model around a gyral surface. (B) Distributions of the vessel diameter for the terminal edges in the present LV2 model and human pial vessel measurements [7]. Note that the measurement data are characterized by the minimum and maximum diameters of the major intracortical vessels, which can be regarded as the terminal points of the pial vessels before penetration into the cerebral cortex.

model exhibits some remarkable features, such as mixtures of large and small vessels, peripheral vessels that suddenly or gradually bifurcate from a large vessel, and large or middle-sized vessels passing across the gyrus.

It can be confirmed that the vessel distribution agrees well with a sketch of the superficial cortical vessels of the human brain (Fig 2 in [7]). Although a straightforward comparison is difficult because of the variation in vascular structures, the present model shows similar tendencies to those observed in actual human vessels [7]: arteries arise from a principal trunk at the sulcus or a hidden origin within the sulcus; a central artery frequently reaches the center region on the gyral surface and divides into numerous sinuous branches; peripheral arteries cover the rest of the gyrus with an angular pathway consisting of a succession of straight edges and angles; the pial venous network is composed of veins larger than the corresponding arteries, including both central and peripheral vessels; the central vein of the gyrus has a star-like pathway; the arterial network covers the venous network on the gyrus with few exceptions. These characteristics might be attributed to some aspects of the present framework: the use of medical information to reconstruct the large vessels and brain surface allows reflection of the overall morphological features of each artery and vein; the multilevel reconstruction leads to different types of vessels such as central or peripheral vessels with various pathways; the region confined reconstruction forms a set of vascular units of arteries and veins within a certain region such as the surface of a cerebral gyrus; adjusting the number of terminal ends on each arterial and venous vascular reconstruction changes the local features of the vascular pathway such as the angular pathway of an artery or the star-like pathway of a vein.

A quantitative agreement of terminal vessel diameters, including their variations, has thus been confirmed. In the present model, the variations in vascular diameter are attributed to the apparent area ratio between the terminal subregion and the total subregions belonging to the same pathway (15). Thus, the diameter depends on the vascular generation process; some pathways might cover many vascular subregions whereas others might not, resulting in a distribution of vascular branch diameters, followed by terminal edges. Although statistical comparisons between the model and actual human vasculature are hard to make because of a few available data in the literature, the present model shows reasonable values, including variation in the terminal edge diameters of the pial vessels, which mainly reflects the data for the pial vessels before they penetrate the cortex. We have also confirmed that the present results agree well in a qualitative manner with other observations of human cortical vessels [35]. The

acceptable agreement with the measurements corroborates our assumption that the vascular pathway and geometry is constructed to supply blood flow to the local surface area as required.

## Morphological structures along vascular pathways

Fig 8A visualizes an example of the arterial pathways from a single root edge with the diameter-defined Strahler order number. The order 1 for terminal edges is increased up to 14 at the root edge going upstream on the pathway, and vascular edges with similar diameters are assigned to the same orders. The relationships between the order number and morphological data are shown in Fig 8B and 8C for arteries and veins, respectively. With the aim of making quantitative comparisons with the measurements, we introduce two different regression lines with respect to the data for $n \in [1, 5]$ and $n \geq 6$ excepting the last order number because of too little data for statistical analysis. It can be observed that the exponential law makes a reasonable fit to the data, with high determination coefficients $R^2$. Although the values for the arteries and veins differ slightly, the overall tendencies are not remarkably different. For the arteries, the slope $b$ given by the present model for the first regression line ($n \in [1, 5]$) and that from the available data for the rat (intra-)cortical vessel [33] is compared in Table 2, and also shown in Fig 8 with the form $\propto n^b$. Note that we could not find available data for human vessels, and therefore we refer to the data for rat vessels.

All the morphological data follow an exponential relationship with the diameter-defined Strahler order, which is known as Horton's law. Here, the slope in the low order is different from that in the higher order. There are three reasons that can be considered for this. First, in

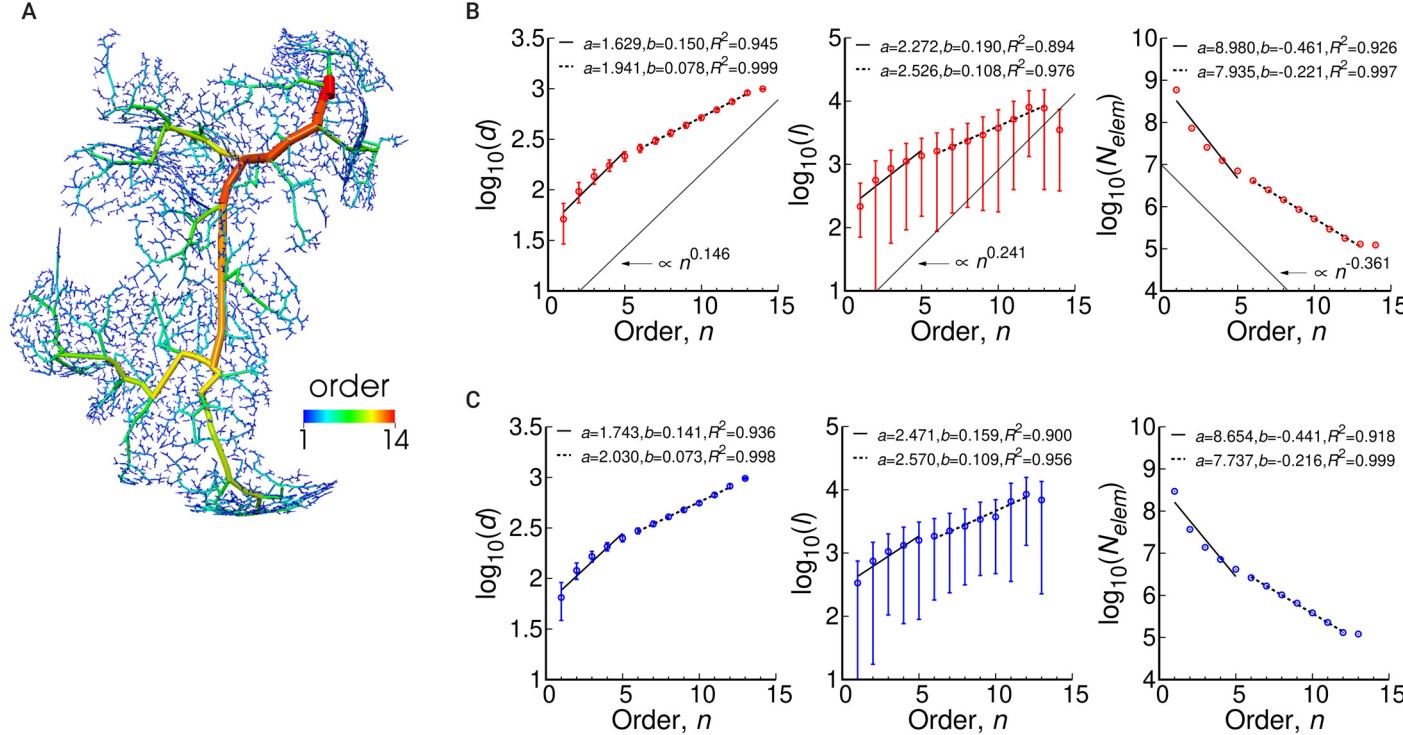

**Fig 8. Morphological features along vascular pathways.** (A) Example of the arterial pathway from a single root with a diameter-defined Strahler order, where the color denotes the order number. (B, C) Morphological features of the diameter, length, and number of elements with respect to the element order for arteries (B) and veins (C). The symbols denote the evaluated values and the solids denote regression lines of the form $\log_{10} q = a + bn$, where two regression lines are fitted for the data from $n \in [1, 5]$ and $n \geq 6$ excepting the last order number. The values of the intercept $a$, slope $b$, and determination coefficient $R^2$ are shown in the graphs. In (B), the fitting curve given by the measurement of rat vessels [33] is also plotted by $\propto n^b$.

**Table 2. The comparison of slope $b$ in the regression line $\log_{10} q = a + bn$ ($n \in [1, 5)$) for morphological features (diameter $d$, length $l$ and number of elements $N_{elem}$) along arterial pathways.**

|  | $d$ | $l$ | $N_{elem}$ |
|---|---|---|---|
| Present ($n \in [1, 5)$) | 0.150 | 0.190 | -0.461 |
| Measurement [33] | 0.146 | 0.241 | -0.361 |

the original idea of the diameter-defined Strahler ordering, the order 0 is assigned to capillary vessels and the order 1 is iteratively assigned according to a diameter distribution; however, in this study, the order 1 is fixed to terminal edges. This gives a large variation in diameter in the order 1, which may lead to different characteristics in the low and high orders. The second reason is a characteristic of the diameter-defined Strahler ordering, which reflects the diameter distribution in each order. We have confirmed that a similar tendency, that is, the order-dependence slope difference, is observed in existing studies using the same ordering (e.g., [32, 36]). It can also be supposed that the original (non-diameter-defined) Strahler ordering would offer a complete linear relationship without any kink with our data. The third reason is that the cerebral vasculature actually has such a two-level fractal feature with respect to the pathway order.

Our slope is in good agreement with the values reported for the rat pial artery [33], where both slopes were obtained by fitting to the data for $n \in [1, 5]$. Note that there are some deviations for the element length and number. The present model connects the branching vertices with a straight line, and does not accept tortuosity without bifurcation or modification of the signed distance function, and thus differences in the element length and number may appear. Nevertheless, the present study is a first attempt to investigate the vascular structure with respect to high branching orders up to 14. Further investigations of real-world data should reveal the fractal nature of the whole-scale cerebrovasculature, and the present outcome may indicate this consideration.

In the present model, no obvious difference in the fractal feature was found between the arterial and venous pathways. To the authors' knowledge, there is no appropriate study of the fractal nature of cerebral venous pathways, and hence, in this regard, we follow an observation on the human pulmonary vasculature [36]. According to this observation, there is little difference in the slope $b$ between the arterial and venous pathways, whereas the intercept $a$ shows clear differences; this is quite similar to the present result. Thus, we can hypothesize that the broad vascular structure and the differences between the arteries and veins, as observed in medical images, are not primary to determine fractal features of vascular pathways.

## Vascular territories of the major cerebral arteries

The vascular territories in the brain hemispheres for the major cerebral arteries (ACA, MCA, and PCA) are shown in Fig 9, where L and R denote the territories for the left and right sides of the brain. Each territory roughly obeys the major cerebral artery, and the territory size varies depending on the arterial type and left or right hemisphere. The area ratios to the PCA are shown in Table 3, where the data for a volume ratio of the vascular territory [37] are also summarized.

The present vascular generation starts from the terminal branches labeled as major cerebral arteries in the image-based model. Thus, although the networks emerge as part of the branching strategy, it can be presumed that the configuration of the image-based model constrains the distribution of the vascular territories. Table 4 shows the number of root edges in the arterial generation of LV1 (i.e., terminal edges of the image-based model). Analogous to the

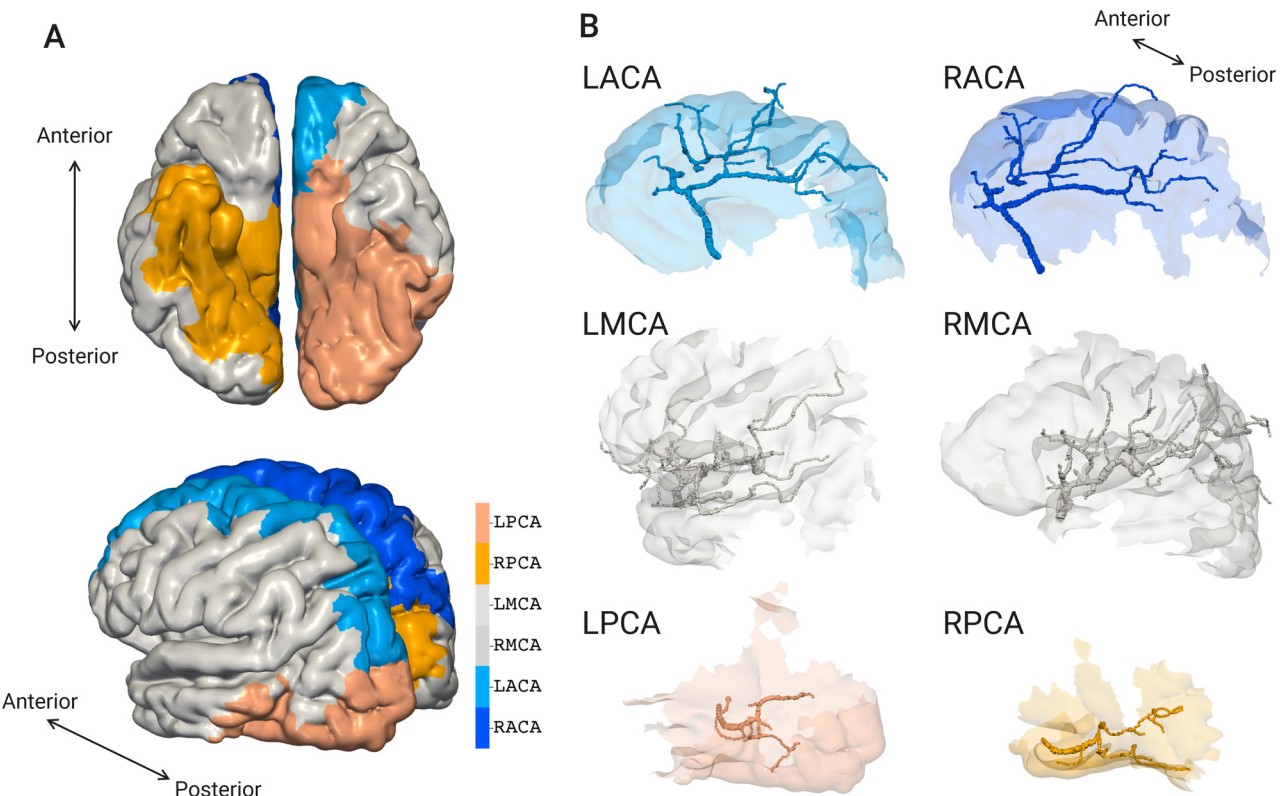

**Fig 9. Vascular territories occupied by major cerebral arteries.** (A) Superimposed views of all the arterial territories. (B) Respective territories with the image-based vascular structures. The vascular territory is classified into anterior cerebral artery (ACA), middle cerebral artery (MCA), and posterior cerebral artery (PCA) for the left(L) and right(R) hemispheres.

territory size in the present model, the number of vessels is ordered MCA>ACA>PCA. However, the ratios to the PCA do not completely correspond with the area ratios. We suppose that other factors also constrain the vascular territories, such as the spatial distribution of large arterial structures.

Although the evaluated territories are for an area in the present model, they have similar values to the volumetric space of the human brain [37]. This supports our presumption that the overall vascular territory follows the macroscale vascular structures in terms of both volume and area. The dependency of territory on macroscale vasculature provides a justification for the present framework, which uses personalized vascular morphology information.

**Table 3. Relative sizes of arterial territories (ACA and MCA to PCA).**

| Territory | Area (present) | Volume [37] |
|---|---|---|
| LACA* | 1.53 | 1.28 |
| RACA* | 1.88 | 1.59 |
| LMCA* | 2.09 | 2.53 |
| RMCA* | 3.27 | 2.99 |

ACA: anterior cerebral artery, MCA: middle cerebral artery, PCA: posterior cerebral artery. L(R): left(right) hemispheres.

**Table 4. The numbers of root arteries with respect to each arterial territory (ACA, MCA and PCA) and their ratios to the PCA.** Here, the total number of terminal edges is 129, as has been already listed in Table 1.

| Territory | Root artery in LV1 | |
|---|---|---|
| | **Num of edges** | **Ratio to PCA** |
| LACA | 18 | 2.25 |
| RACA | 18 | 1.64 |
| LMCA | 33 | 4.13 |
| RMCA | 41 | 3.73 |
| LPCA | 8 | – |
| RPCA | 11 | – |

Even though the territories roughly follow the configurations of the image-based model, they are indeterministic and unpredictable, because the final configuration is given through the generation process in the modeling. Our preliminary examples have shown that the territories vary with changes to the configuration of the vascular subregion, although the magnitude relation among arterial territories is not altered. This variability in the results for the arterial territories is an interesting feature from the viewpoints of both modeling and physiology. It has been argued that territory patterns are not constant, because of not only interindividual hemodynamic circumstances, but also intraindividual and time-related variabilities in the alteration of hemodynamic circumstances [38]. Furthermore, it has been shown that anastomoses between the arterial territories play an important role in collateral blood flows [39], which may relate to alteration of the demarcation between arterial territories. A quantitative evaluation of territorial variability is beyond the scope of this study; nevertheless, numerical experiments will elucidate essential components of the variability of vascular territories by allowing analysis of different configurations based on different personal data.

## Effects of coarse-scale vascular subregions on vascular structures

We compared vascular structures using different numbers of coarse-scale vascular subregions, $N_u \equiv N_d^{(LV1)} = 1000,\ 2000$ and $4000$.

Fig 10A shows arterial structures from a single root as an example, where the vessels are individually visualized for the different reconstruction levels. As described above, coarse-scale vascular structures are first generated in the LV1 reconstruction, followed by the generation of fine-scale vascular structures in LV2. Although this trend is commonly observed in all the models, the coarse-scale degree differs according to the parameter $N_u$. With a decrease in $N_u$, each coarse-scale subregion area becomes larger, and thus the coarse-scale vascular model consisting of large-size vessels must cover a broader region of the brain surface.

To evaluate how such a hierarchical difference appearing in the modeling influences consequent vascular pathways and geometries, we investigated pathway features as shown in Fig 8. Contrary to expectations, there was no remarkable difference in the parameters of Horton's law among the models using different $N_u$.

In addition, we evaluated the path length $C$. Fig 10B shows that although we could find a slight difference in the frequency around $60 \le C \le 120$, where the condition of $N_u = 1000$ increases the probability of generating a larger path length ($\sim 80$ mm), there is no obvious difference in the overall distribution.

To investigate the anatomical relationship between the coarse-scale vasculature in the present model and the actual cerebral vasculature, we evaluated the distribution of terminal edge diameters for the LV1 model and compared them with the measurement data for the human

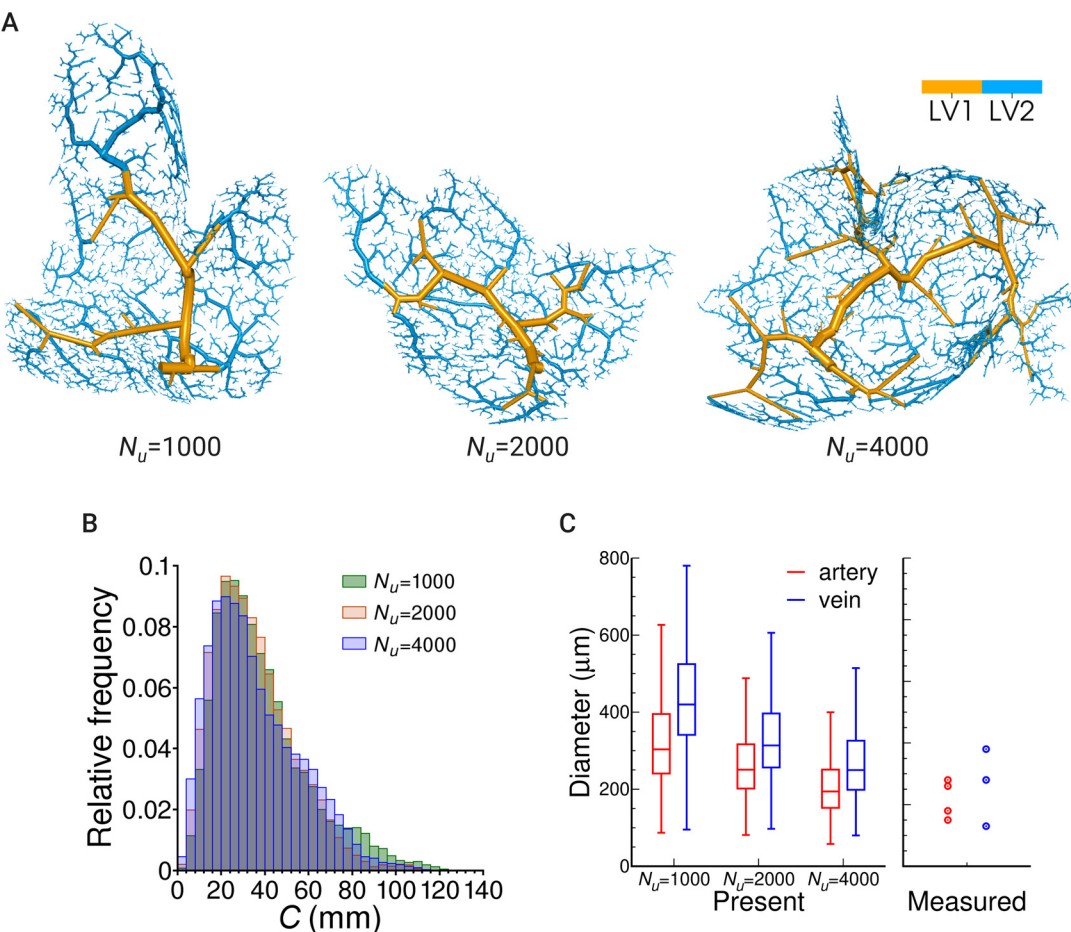

**Fig 10. A relationship between vascular pathways and geometries and the number of coarse-scale vascular subregions.** (A) An example of arterial structures from a single root. The color differentiates the reconstruction level of the vascular generation. (B) The relative frequency of the path length $C$, the summation of length for all the edges consisting of the whole pathway from the root to terminal edges. (C) The distribution of terminal edge diameters for the LV1 model and comparison with the measurement data for the human cortical vessels [7]. The measurement data are referred to as the minimum and maximum diameters of central and peripheral vessels.

cortical vessels [7] (Fig 10C). Here, the measurement data is referred to using the diameters of the central and peripheral vessels of the superficial cortical vessels as a coarse-scale vasculature, with these being relatively larger than the diameters shown in Fig 7. With an increase in $N_u$ the diameters decrease, and the values approach the measurements for these parameters. This indicates that morphological features of the vasculature can be explicitly modeled at an anatomical level.

Unfortunately, we could not provide a proper parameter for $N_u$ at this time, because there is no physiological evidence to determine the configuration of the vascular subregions. In other words, the model has the flexibility to provide multiple situations of blood flow simulation, and the relationship between local and entire flows can be controlled by varying the coarse-scale subregion parameter $N_u$.

We must emphasize that the coarse-scale subregion parameter $N_u$ influences the computational time of the modeling (Table 5). The reconstruction is carried out using a single CPU-core (Intel Xeon E5-2600 v3 processor). The major computational cost occurs in searching for

**Table 5. Comparison of execution time for the reconstruction at different $N_u$.**

| $N_u$ | Execution time (s) * | | |
|---|---|---|---|
| | LV1(artery/vein) | LV2(artery/vein) | total |
| 1000 | 70/27 | 2931/611 | 3639 |
| 2000 | 434/71 | 1029/246 | 1780 |
| 4000 | 3397/427 | 483/258 | 4565 |

* Note that the current code is not optimized sufficiently, and thus the execution time is possibly to be reduced.

the closest edge when adding a new terminal vertex (note that in our coding, it is reduced by utilizing background voxel meshes). In the LV1 process, as $N_u$ increases, the number of vascular edges also increases, which results in a non-linear increase in the cost and time required to find the closest edge. The vascular generation in the LV2 process is completely independent within each LV1-subregion, and thus a computational cost occurs as a result of the multiplication of the number of LV2-subregions and the cost per LV2-subregion. Therefore, the non-monotonic behavior between the computational time and parameter $N_u$ appears in the total reconstruction. If we employ larger values for the parameter $N_u$ (using the same number of vascular subregions in total), the computational time becomes huge. The multilevel reconstruction has the additional advantage that it circumvents such a huge increase in total computational time. We also remark that a parallel computing technology promises to reduce the modeling time because of the independence of the LV2 reconstruction, and this could be enhanced with a smaller $N_u$.

## Influence of brain shape on vascular structures

We discuss here how brain shape affects the vascular structure, and show the importance of using real-world geometry. We create a *simplified* model for the shape of the hemispheres *actual* by artificially eliminating major folds (cerebral sulcus) in the brain, as shown in Fig 11A. Thus, the overall shape is similar to the actual shape, while the local geometry is not. The surface area of the simplified model is 73270 mm², which is approximately 0.72 times smaller than that of the actual model, due to elimination of the cerebral sulci. All the parameters are set to be the same as in Table 1, except for the surface area of the hemispheres, resulting in different number-densities for the terminal ends to those used in the actual model. The same number of total terminal edges are employed so that the graph characteristics in both the actual and simplified models are equivalent.

The superficial vessels follow the concave surface of the simplified model (Fig 11B). The relationships between the Euclidean distance $L$ and path length $C$ between the root vertex and terminal ends for the whole-pathways are shown in Fig 11C. Here, a linear regression line is also plotted for each model. A proportional relationship holds between $C$ and $L$ in both the actual and simplified models, although the slopes are different. The relative frequencies of $C$ and $L$ are shown in Fig 11D and 11E, respectively. The path length $C$ for the actual model tends to provide slightly larger values, whereas no remarkable difference is observed for the Euclidean distance $L$ between the models. Furthermore, there is no obvious difference in the number of bifurcations on the whole pathways of the actual and simplified models (Fig 11F). Even under these circumstances, the tortuosity $C/L - 1$ shows a considerable difference in the relative frequency (Fig 11G). The data are fitted to logarithmic normal distributions. The actual model tends to provide larger degrees of tortuosity in the vascular pathways.

The simplified model roughly duplicates the actual brain shape, with the elimination of the cortical folding pattern. Moreover, we set the parameters to be the same, except for the surface

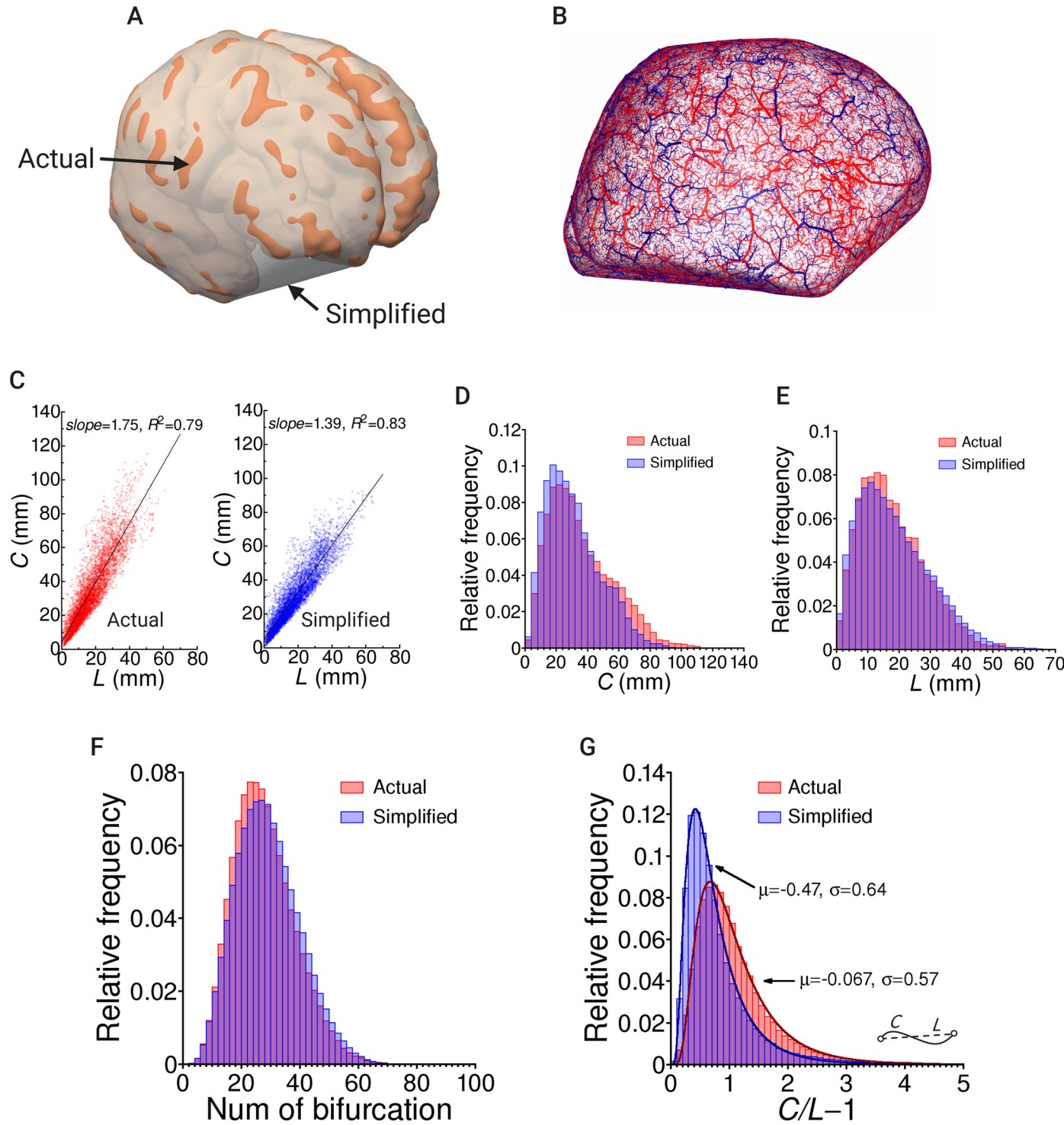

**Fig 11. Comparisons of statistical features for vascular structures of whole-pathways in the models reconstructed with actual and simplified brain hemisphere shapes.** (A) Hemisphere shapes for the actual model (orange) and simplified model (white), in which the major folds (cerebral sulcus) were artificially eliminated. (B) Reconstructed vessels in the simplified model for arteries (red) and veins (blue). (C) Relationships between the Euclidean distance $L$ and path length $C$ of the whole pathways between the root vertex and terminal ends. A linear regression line is also plotted for each model. (D,E) The relative frequencies of the path length $C$ (D) and Euclidean distance $E$ (E) for the whole pathways. (F) Relative frequencies of the number of bifurcations on the whole pathways. (G) Relative frequencies of the index of whole-pathway tortuosity given as $C/L - 1$, where curves fitted using logarithmic normal distributions are shown.

area in the modeling. Thus, the basic vascular structure of the simplified model follows that of the actual model. We have confirmed that the pathway characteristics based on Horton's law do not vary between the models. Therefore, we again confirm that the pathway characteristics are not sensitive to the brain shape.

Nevertheless, some considerable differences arise in the whole-level pathways. The path length $C$ becomes longer, and we could also find a similar tendency for the Euclidean distance $L$ around the peak ($\sim 15$ mm), although it is not obvious. This is not surprising, because the surface area differs between the models. It is easy to suppose that the larger surface area provides larger $C$ with the same number of terminal ends.

The important thing is that the models provide different magnitudes for the non-dimensional relationship between $C$ and $L$. The slope of the linear regression line between $C$ and $L$ shows a larger value in the actual model, and a considerable difference can be seen in the distribution of the ratio defined as the tortuosity index. This indicates that the whole pathways from upstream to downstream on the brain surface (for the arteries, or vice versa: downstream to upstream for the veins) obey the local curvature of the cortical folds, and that the associated blood flows undergo frictional resistance in the tortuous pathways. This is also an important reason to employ real-world geometry in the modeling, to appropriately address the three-dimensional spatial distribution of the cerebrovasculature for the blood flow simulations.

## Model limitations and future considerations

In the present formulation, the vascular generation in a certain level only reflects the terminal edges in the former level as the root edges. Additionally, the arterial generation does not take into account the midstream pathway and geometry of the vein. This admits overlapping vascular structures among the different reconstruction levels and types of vessels. It has been found that the venous network is disposed on the cerebral surface, and that the cortical arteries cross over the veins [7, 35]. This arrangement could change the vascular pathway, and therefore comparisons with or without a constraint considering the existing vasculature will be needed.

There are many anastomoses among the arteries and veins of the human cortical vessels [7], and the roles of anastomoses have been discussed under the condition of arterial occlusions [39, 40]. The present formulation may be able to address several types of anastomoses based on information on vascular subregions, either on-line or post-reconstruction. For instance, by connecting the pathways entering different vascular subregions, we can model a set of anastomoses to produce a vasculature that broadly spreads over the brain surface.

In this study, the geometries of the brain hemispheres and macroscale vasculatures were extracted from different subjects according to data accessibility. This might result in the loss of information regarding personal specificity, and therefore further modeling using the same subject-specific geometries is desirable. Moreover, the personalized geometries were extracted from different imaging modalities (MRI and CT). Although we manually adjusted the positions, a sophisticated position adjustment algorithm, such as nonrigid multimodal registration, is required to ensure generality and avoid systematic errors in the modeling. In addition, comparisons among the models with different subject-specific data are necessary to understand the commonalities and variabilities of cerebrovascular systems.

## Concluding remarks

We have provided a multiscale model of the human cerebrovasculature using a novel framework based on a hybrid combining image-based geometries and a mathematical algorithm. The resulting MRC algorithm enables the generation of tree structures representing arterial and venous systems in a multilevel manner according to the brain shape and large vessels

obtained from medical images. A set of vascular units corresponding to the arterial and venous systems is then synthesized. The model was applied to the reconstruction of superficial cortical vessels as the mesoscale vasculature linking the micro and macroscale vasculatures. We validated the model by comparing the geometrical features obtained with those of actual cortical vessels, and some remarkable features and hypotheses were derived: (i) vascular pathways form to ensure a reasonable supply of blood to the local surface area; (ii) the fractal features of vascular pathways are not sensitive to the overall or local brain geometries; and (iii) whole pathways connecting the upstream and downstream entire-scale cerebral circulation are highly dependent on the local curvature of the cerebral sulci.

Although the current model does not involve a microvascular system, we believe the present framework will provide a way to generate a complete set of cerebrovascular structures by combining it with an existing microvascular model. Flow simulations using the full-scale cerebrovascular model will reveal many physical aspects of the cerebral blood flow.

## Supporting information

**S1 Supplement. Validation of the vascular generation algorithm.**
(PDF)

## Acknowledgments

The authors would like to thank Mr. Hisashi Kato, Dr. Tomohiro Otani and Dr. Luosha Xiao for the reconstruction of vessel structures from medical images, and also Dr. Kenichiro Koshiyama for fruitful discussions on the tree generation algorithm and Dr. Naoki Takeishi for insightful comments on the overall view of the modeling.

## Author Contributions

**Conceptualization:** Satoshi Ii, Shigeo Wada.

**Data curation:** Hiroki Kitade, Yoshiyuki Watanabe.

**Formal analysis:** Satoshi Ii.

**Funding acquisition:** Shigeo Wada.

**Investigation:** Satoshi Ii, Hiroki Kitade.

**Methodology:** Satoshi Ii, Hiroki Kitade, Shunichi Ishida, Yohsuke Imai, Shigeo Wada.

**Project administration:** Shigeo Wada.

**Software:** Hiroki Kitade.

**Supervision:** Satoshi Ii, Shigeo Wada.

**Validation:** Satoshi Ii, Hiroki Kitade.

**Visualization:** Satoshi Ii.

**Writing – original draft:** Satoshi Ii, Hiroki Kitade.

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
