## [Decision Letter · Decision Letter 0]

16 Nov 2019

Dear Dr Ii,

Thank you very much for submitting your manuscript 'Hybrid modeling of whole-scale cerebrovasculature based on personalized morphometry and mathematical algorithm' for review by PLOS Computational Biology. Your manuscript has been fully evaluated by the PLOS Computational Biology editorial team and in this case also by independent peer reviewers. The reviewers appreciated the attention to an important problem, but raised some substantial concerns about the manuscript as it currently stands. While your manuscript cannot be accepted in its present form, we are willing to consider a revised version in which the issues raised by the reviewers have been adequately addressed. We cannot, of course, promise publication at that time.

Sincerely,

Daniel A Beard

Deputy Editor

PLOS Computational Biology

Daniel Beard

Deputy Editor

PLOS Computational Biology

[LINK]

Reviewer's Responses to Questions

**Comments to the Authors:**

Reviewer #1: See attached file

Reviewer #2: The review is uploaded as an attachment.

**Have all data underlying the figures and results presented in the manuscript been provided?**

Reviewer #1: Yes

Reviewer #2: No: Either these data have not been made available or I haven't been able to find them.

It is stated that: "All relevant data are within the manuscript and its Supporting Information files."

These are only pdf files and contain no data, unless the data is simply considered irrelevant here.

This concerns both large-scale datasets such as imaging CT and MRI pre- or post-processed data or reconstructed volumes, and summary statistics.

PLOS authors have the option to publish the peer review history of their article (what does this mean?). If published, this will include your full peer review and any attached files.

Reviewer #1: No

Reviewer #2: No

---

## [Decision Letter · Decision Letter 1]

1 Jan 0001

Dear Dr Ii,

Thank you very much for submitting your manuscript "Modeling of multiscale human cerebrovasculature: a hybrid approach using image-based geometry and mathematical algorithm" for consideration at PLOS Computational Biology.

As with all papers reviewed by the journal, your manuscript was reviewed by members of the editorial board and by several independent reviewers. In light of the reviews (below this email), we would like to invite the resubmission of a significantly-revised version that takes into account the reviewers' comments.

We cannot make any decision about publication until we have seen the revised manuscript and your response to the reviewers' comments. Your revised manuscript is also likely to be sent to reviewers for further evaluation.

Sincerely,

Daniel A Beard

Deputy Editor

PLOS Computational Biology

Daniel Beard

Deputy Editor

PLOS Computational Biology

Reviewer's Responses to Questions

**Comments to the Authors:**

Reviewer #1: Authors have significantly improved the manuscript, the manuscript has a number of good components, but it still lacks a clear description of the study presented. As far as I see the main message in this manuscript is Figure 5 illustrating that it is possible to reconstruct the cerebrovascular network, in which the large vessels are informed by imaging data while the smaller vessels are generated using approximation techniques.

I think this paper should be presented as a method paper discussing how best to approximate the small vessels not captured by the imaging data.

To convey this story I suggest to include the following changes:

Abstract: Remove the first paragraph - start by noting that cerebral vasculature is complex and with standard imaging techniques it is difficult to segment all vessels in the network, and that this study proposes a method to obtain these networks by explicitly segmenting the large vessels combined with approximation techniques for small vessels. The next few sentences should address how the method proposed here is validated, and what are its limitations.

Introduction: Relate the method proposed here to other segmentation studies extracting networks from imaging data. Motivate what you intent to do with the segmented network - compute flow, couple it with a model for cerebral autoregulation, etc etc. But this would be future work as it is not included in the current study.

Methods: I do not like the word "modeling" as this is really "network reconstruction". I suggest to take out the word modeling instead generating a title e.g. "Network Generation". Moreover, I suggest removing the first paragraph page 4 lines (88-102)

Close to figure 2 - add more details describing how the image was segmented and the network generated for the large vessels. Include a description of how the vessels in the network are extracted from the image and connected. Also describe what quantities are stored , e.g. vessel length, vessel diameter?

Page 4:Rename the section on "preparation for mathematical modeling" the title does not describe what is "modelleld"

I think the authors are describing that the next step needed to generate a network is to extract the volume (space) in which the network lies. Therefore, use a title e.g. Extracting Brain Volume (or volume enclosed by the network).

Page 5: Delete section "Inputs to mathematical modeling"

Page 5: Change title "Mathematical model for mesocale vasculature" to "Construction of mesocale part of the network" or something like that.

Page 5: Here you discuss "potential features" focus on addressing features included in this study, e.g. you do not show any flow computations so remove lines 167-168

Page 5: Title "Definition of vascular pathways and structures" change this to generation of vascular pathways and structures.

Page 7: Authors discuss how the vessel radii are evaluated by studying flow in the network, but no simulations are provided showing the flow. Either take the flow discussion out or show flow calculations.

Page 10: Should the title of this section be "Network Validation" or something along these lines. For each evaluation type compare predicted network to data. Currently this section reads as you are calculating several metrics, but I don't see how they are related to data.

Page 11: Separate the results (showing the generated networks and how they compare to data) and discussion addressing how the technique presented here compare with other studies.

See e.g. the study by Meijs et al. Scientific Reports 2017 DOI:10.1038/s41598-017-15617-w

or

Reviewer #2: I would like to thank the authors, who made a substantial effort to improve their paper and to respond to the reviewers' comments.

Indeed, most of the revisions and significant issues raised appear to have been reasonably addressed.

However, I encourage the authors to carefully review the manuscript again, as I have identified a number of minor errors (typographical, spelling or grammatical) that would require attention, especially in the added paragraphs.

Also, even though it may seem like a detail, I would like to point out one formulation issue that caught my attention, as it appears several times in the manuscript, including in the title: "multiscale model(ing) of human cerebrovasculature" would be more correct than "model(ing) of multiscale human cerebrovasculature", in my opinion.

**Have all data underlying the figures and results presented in the manuscript been provided?**

Reviewer #1: Yes

Reviewer #2: Yes

PLOS authors have the option to publish the peer review history of their article (what does this mean?). If published, this will include your full peer review and any attached files.

Reviewer #1: No

Reviewer #2: No
---

## [Decision Letter · Decision Letter 2]

11 May 2020

Dear Dr Ii,

We are pleased to inform you that your manuscript 'Multiscale modeling of human cerebrovasculature: a hybrid approach using image-based geometry and a mathematical algorithm' has been provisionally accepted for publication in PLOS Computational Biology.

Best regards,

Daniel A Beard

Deputy Editor

PLOS Computational Biology

Daniel Beard

Deputy Editor

PLOS Computational Biology

Reviewer's Responses to Questions

**Comments to the Authors: **

Reviewer #1: No further comments

**Have all data underlying the figures and results presented in the manuscript been provided?**

Reviewer #1: Yes

PLOS authors have the option to publish the peer review history of their article (what does this mean?). If published, this will include your full peer review and any attached files.

Reviewer #1: No

---

## [Editor Report · Acceptance letter]

9 Jun 2020

PCOMPBIOL-D-19-01659R2 

Multiscale modeling of human cerebrovasculature: a hybrid approach using image-based geometry and a mathematical algorithm

Dear Dr Ii,

I am pleased to inform you that your manuscript has been formally accepted for publication in PLOS Computational Biology. Your manuscript is now with our production department and you will be notified of the publication date in due course.

With kind regards,

Sarah Hammond
